# Quantifying the effects of environmental factors on wildfire burned area in South Central US using integrated machine learning techniques

Sally S.-C. Wang[1], Yuxuan Wang[1]

[1]Department of Earth and Atmospheric Sciences, University of Houston, Houston, Texas 77024, USA

*Correspondence to*: Yuxuan Wang (ywang246@central.uh.edu)

**Abstract.** Occurrences of devastating wildfires have been increasing in the United States for the past decades. While some environmental controls, including weather, climate, and fuels, are known to play important roles in controlling wildfires, the interrelationships between these factors and wildfires are highly complex and may not be well represented by traditional parametric regressions. Here we develop a model consisting of multiple machine learning algorithms to predict 0.5º×0.5º-gridded monthly wildfire burned area over the South Central United States during 2002-2015 and then use this model to identify the relative importance of the environmental drivers on the burned area for both the winter-spring and summer fire seasons of that region. The developed model alleviates the issue of unevenly-distributed burned area data, predicts burned grids with Area Under the Curve (AUC) of 0.82 and 0.83 for the two seasons, and achieves temporal correlations larger than 0.5 for more than 70% of the grids and spatial correlations larger than 0.5 (p<0.01) for more than 60% of the months. For the total burned area over the study domain, the model can explain 50% and 79% of the observed interannual variability for the winter-spring and summer fire season, respectively. Variable importance measures indicate that relative humidity (RH) anomalies and preceding months' drought severity are the two most important predictor variables controlling the spatial and temporal variation of gridded burned area for both fire seasons. The model represents the effect of climate variability by climate-anomaly variables and these variables are found to contribute the most to the magnitude of the total burned area across the whole domain for both fire seasons. In addition, antecedent fuel amounts and conditions are found to outweigh the weather effects on the amount of total burned area in the winter-spring fire season, while fire weather is more important for the summer fire season likely due to relatively-sufficient vegetation in this season.

## 1. Introduction

Wildfire is an important process maintaining the balance of terrestrial ecosystems. Wildfire occurrence is controlled by a complex interaction among fuel, weather, and climate (Bowman et al., 2009; Pausas and Keeley, 2009). In recent decades, many regions of the world have experienced an increase in frequency and intensity of wildfires, which may be possibly connected to changes in regional climate (Balshi et al., 2009; Barbero et al., 2015; Carvalho et al., 2008; Flannigan et al., 2009; Westerling et al., 2006; Westerling, 2016). More intense and more frequent wildfire activities not only heighten ecosystem

vulnerability but also cause poor air quality (Jaffe et al., 2008; Pellegrini et al., 2017; Wang et al., 2018; Yue et al., 2015). Thus, it is imperative to understand how wildfires would respond to changes in environmental factors in a warming climate.

Previous studies revealed the importance of several environmental factors on wildfires. Fuel availability and composition across regions can affect fire developments such as fire likelihood and spread efficiency (Nunes et al., 2005; Parks et al., 2012). Weather influences fuel moisture by changing precipitation and humidity and controls fire spread through winds.

Long-term climate change can alter both fuel and weather conditions, for example by adjusting vegetation distributions and the frequency of fire-favorable atmospheric conditions (Heyerdahl et al., 2008; Keyser and Westerling, 2017; Morgan et al., 2008; Zubkova et al., 2019), therefore changing fire regimes. Past studies also highlighted that the complex interplay between fuel, weather, climate, and wildfires can vary depending on spatial scale, fire size, region, and season. For instance, the relationships between fire activity and the environmental controls can exhibit complex nonlinearities across the spatial scale

gradient (Peters et al., 2004). Fuel and topography mainly regulate fires at a local scale, while weather and climate control fires at a broad spatial scale (Parks et al., 2012). In terms of fire size, it was found that the major controlling factors could shift from fuel and topography to weather as fire size increases in boreal forests (Liu et al., 2013; Fang et al., 2015). In the western Mediterranean Basin where land heterogeneity is large, influences of fuel can outweigh influences of climate and weather on large fires (Fernandes et al., 2016). Therefore, it is challenging to examine the relative importance of the environmental drivers

on wildfires due to the complex interrelationships among them.

One common method to explain the relationships between fire regimes (e.g. fire sizes or fire occurrences) and environmental factors is regression. This method is also used to evaluate the relative importance of different environmental controls (Littell et al., 2009; Slocum et al., 2010; Parisien et al., 2011; Yue et al., 2013; Liu & Wimberly, 2015; Fernandes et al., 2016). Among a wide range of regression techniques used, non-parametric machine learning algorithms have emerged as

an important tool to predict wildfires because they rely on fewer pre-assumptions about the data. Bedia et al. (2014) used non-parametric multivariate adaptive regression splines (MARS) to model the monthly burned area for the phytoclimatic zones in Spain of sizes ranging from 25 km x 25 km to 100 km x 100 km. Amatulli et al. (2013) used two machine learning approaches, Random Forest (RF) and MARS, to estimate monthly burned area in five countries in Europe with a spatial resolution ranging from 300 km x 300 km to 1000 km x 1000 km. In these studies, the machine learning methods were used to estimate total

burned area aggregated over a large-scale domain, e.g. on an ecoregion or a country scale (Table S1). However, fewer studies have explored the utility of machine-learning methods in resolving the within-domain and grid-level relationships between fires and the environmental drivers. A particular challenge in predicting burned area of fires at the grid level across a broad region relates to the uneven distribution of burned area both spatially and temporally, where the number of grids of large burned area is much smaller than the number of those with small or zero burned areas. For example, Steel et al. (2015) showed

that for fires in California, small fires (< 25 ha each) contributed to 87% of the total number of grids burned but only 17% of the total burned area, whereas large fires (> 150 ha each) accounted for only 3% of the total number of burned grids but made up 64% of the total burned area. Thus, at the grid level the majority class is non-burn wildlands or small fires, while the

minority class is large fires. As most data-driven regression algorithms, parametric or non-parametric, would favor the majority class, large fires will be underpredicted for grid-level predictions.

In this study, we develop a model consisting of multiple machine learning techniques to predict wildfire burned area at the grid level over the vegetation-rich and thus fire-prone region of the South Central United States (US), which encompasses four states -Texas, Oklahoma, Louisiana, and Arkansas – as shown in Figure 1. The study region is chosen for several reasons. First, this region is composed of similar vegetations which are plains and oak-hickory forests. Second, the vegetation-rich region of the South Central US is fire-prone and has experienced periodically large wildfires in recent years, such as the 2011

Texas fires (Long et al., 2013; Nielsen-Gammon, 2012), but the region as a whole has been much less studied compared to the western US. Third, this region is projected to have the highest risk of wildfires in 2031-2050 across the continental US (An et al., 2015; Fann et al., 2018). In terms of the prediction method, the integrated machine learning model aims at mitigating the problem of uneven distribution of burned area data and improving the accuracy of predicting wildfire burned area at a grid-scale of 0.5° x 0.5°. Using the prediction model developed here, the goal of this paper is to estimate the relative importance of

different environmental factors on wildfire burned area in the study region which would be useful for future fire prediction as well as understanding the linkage between wildfires and climate change.

     The study period is from 2002 to 2015. For each year, we predict gridded wildfire burned area at the monthly scale for the typical bimodal wildfire seasons over the region (Figure S1): the winter-spring fire season from January to April and summer fire season from July to September (Zhang et al., 2014). Wildfires during the winter-spring wildfire season are

typically associated with dry and strong winds resulting from the large-scale low-pressure systems (Heilman et al., 1998; Jones et al., 2013), while wildfires in the summer are mostly driven by the abundance of dry or dead vegetations produced from the dry season (Jones et al., 2013). These two seasons contribute 76% of the annual total burned area, indicating that natural environmental conditions in these months are most conducive for wildfires. While wildfires do occur outside the fire seasons, their lower frequency implies that non-natural factors (e.g. human actions) can be relatively more important. As our study does

not focus on human factors, we choose to exclude other months of the year.

     The rest of the paper is organized as follows: Section 2 introduces data incorporated into the model. Section 3 describes the developed model and validation method. Section 4 presents the results of model validation and evaluation. In section 5, we analyze the relative importance of individual variables and the environmental controls at different spatial scales. Discussion and conclusion are given in section 6.


## 2. Data

### 2.1 Wildfire burned area

     The model predicts wildfire burned area at a grid-scale of 0.5°×0.5° over the study region. Wildfire burned area is chosen as the target variable because it is a widely-used parameter for quantitative assessment of fire danger and fire impact

(Amatulli et al., 2013; Balshi et al., 2009; Yue et al., 2013). Wildfire information over the study period (2002-2015) is obtained from the Fire Program Analysis Fire-Occurrence Database (FPA-FOD). The FPA-FOD collects daily wildfire reports from federal, state, tribal, and local governments. The dataset includes wildfire burned area, fire location in longitude and latitude, and fire discovery date from 1992 to 2015 (Short, 2017). The FPA-FOD fire data excludes prescribed fires except for the prescribed fires that escape their planned perimeters and become wildfires. A known caveat of this database is that it does not include some small fires that occur on private lands. Short (2014) reported that for the period of 1992-1997 the national total number of wildfires from the FPA-FOD is about 30% lower compared to that from the US Department of Agriculture Forest Service (USFS) Wildfire Statistics, although the national total burned area is consistent between the two datasets. Thus, our model will not be able to predict those small fires missing from the FDA-FOD as such information is not in the training dataset.

The FPA-FOD wildfire data is point data at a daily time step. As the prediction model deals with the monthly total burned area at a spatial resolution of 0.5°×0.5°, we aggregate the daily point burned area into 0.5°×0.5° grid cells based on fire longitude and latitude and sum the burned area in each grid by month. The resulting dataset of monthly burned area has nearly 70% of the grids with burned area less than 10 ha or non-burned. To reduce skewness and improve data symmetry, we apply the log transformation function $ln(x+1)$, where x is the gridded monthly total burned area. The log-transformed burned area is the target variable of the model.

## 2.2 Predictor variables

Based on previously published studies, we collect a number of predictor variables that are thought to influence wildfire burned area (Fang et al., 2015; Keyser and Westerling, 2017; Liu and Wimberly, 2015; Riley et al., 2013; Yue et al., 2013) and group them into four categories of environmental controls (Table 1): weather, climate, fuel, and fixed-geospatial variables. These predictor variables are listed in Table 1 and described below. All the variables, including continuous and discrete thematic variables, are resampled to a spatial resolution of 0.5°× 0.5° by the nearest neighbor resampling method (Baboo and Devi, 2010). The nearest neighbor resampling method assigns a value to the new grid according to the value of the original grid closest to the center of the new grid. The resampling method has the advantages of being efficient and not changing any value from the original dataset.

### 2.2.1 Weather variables

The meteorological data are obtained from the North American Regional Reanalysis (NARR) with a spatial resolution of 32 km x 32 km (Mesinger et al., 2006). The weather variables include the monthly total accumulated precipitation and the monthly means of the following variables: daily precipitation, daily average and maximum temperature, zonal (U) and meridional (V) components of wind at 10 m, and daily average and minimum relative humidity (RH). In order to select extreme conditions that are likely to induce wildfires on a sub-monthly time scale, we also include the number of consecutive days without rainfall within a month, which is based on daily precipitation from the NARR data. Another weather pattern

conducive for wildfires is drought (Gudmundsson et al., 2014; Riley et al., 2013; Turco et al., 2017). Drought depicts the extreme condition of water deficit in the coupled land-atmosphere system that can be driven not only by lack of precipitation but also by excessive evaporation. We use the Standard Precipitation and Evaporation Index (SPEI) to represent drought intensity (Vicente-Serrano et al., 2009). The SPEI incorporates both precipitation and potential evapotranspiration to estimate climatic water balance at different time scales (1 to 48 months). In this study, we use the 1-month SPEI from the global SPEI database (http://spei.csic.es/database.html) with a spatial resolution of 0.5°× 0.5°. Positive values of SPEI represent wetter than normal conditions and negative values indicate conditions that are drier than normal.

Weather conditions in the preceding months are also known to influence fire development. For example, an increase of precipitation in the preceding months can promote biomass growth and provide fuels for a widespread of larger wildfires in a later month (Fréjaville and Curt, 2017; Littell et al., 2009). To consider such lagged effects, for a given month $t$, we calculate the averages of the aforementioned weather variables from the months $t$-1 to $t$-12. We then include those lagged variables that have correlation coefficients (r) larger than 0.5 with wildfire burned area of month $t$ but are not strongly correlated with the same variables of month $t$ (r < 0.5). For the winter-spring fire season, the antecedent variables that pass this criterion are the monthly mean of daily precipitation of months $t$-1 and the average SPEI of the months $t$-1, $t$-1 to $t$-2, $t$-1 to $t$-3, $t$-1 to $t$-4, $t$-1 to $t$-5, and $t$-1 to $t$-6. For the summer fire season, the selected antecedent variables are the average of monthly mean temperature for months $t$-1 and $t$-1 to $t$-2, monthly mean of daily precipitation for months $t$-1, $t$-1 to $t$-2 and $t$-1 to $t$-3, and mean SPEI of months $t$-1, $t$-1 to $t$-2, and $t$-1 to $t$-3.

**2.2.2 Climate variables**

Inputs of climate variables to the model include both climate anomalies and 22-year (1979-2000) means and standard deviations of selected meteorological variables from the NARR data. Here climate anomalies refer to the departure of monthly mean meteorological variables from their long-term averages over 1979-2000, thereby representing the effects of climate on meteorological conditions. The climate anomalies are calculated for the monthly total precipitation and monthly means of daily average precipitation, daily average and maximum temperature, average and minimum RH. The long-term average and standard deviation of meteorological variables characterize the spatial and temporal patterns of the mean climate conditions, which can determine the typical vegetation of the study region and hence influence fire occurrence and size (Keyser and Westerling, 2017). We use the 22-year means and standard deviations of monthly total accumulated precipitation and monthly means of daily average and maximum temperature, and daily average precipitation. As climatological means and standard deviations do not vary with time, they are grouped with the geospatial variables later in the study as the category of fixed variables.

### 2.2.3 Fuel variables

Fuel variables are selected to estimate the fuel effect on burned area and these variables include monthly mean of Leaf Area Index (LAI), sum of neighboring LAI, and soil moisture. The LAI is the ratio of the total one-sided area of green leaf area per unit ground surface area, which has been widely used to describe the structural property of a plant canopy (Watson, 1947; Chen and Black, 1992). Additionally, LAI is correlated with important metrics of canopy fuel loads, such as canopy bulk density (Keane et al., 2005; Steele-Feldman et al., 2006). The monthly mean LAI at a spatial resolution of 500 m is obtained from MODerate resolution Imaging Spectroradiometer (MODIS) instruments (Myneni et al., 2015). Besides local LAI values, to capture the effects of spatial autocorrelations, we consider each grid cell as the center of a 3-by-3 grid matrix and compute the summation of the LAI from the center grid's eight neighboring grids. This summation is referred to as the 'sum of neighboring LAI' and included as a predictor variable. The lagged effects of fuel buildup in the preceding months are expected to influence wildfire occurrence and size. Using the same criteria to select antecedent weather variables (section 2.2.1), the averages of LAI and sum of neighboring LAI for the months $t-1$ to $t-6$ are selected as antecedent fuel variables for the winter-spring fire season, but no such variables are included for the summer fire season because none passes the selection criteria.

Fuel moisture is a critical property for evaluating fire danger. As fuel moisture data is limited, soil moisture is often used as an indicator of fuel moisture because of the strong correlation between the two (Krueger et al., 2016). Here, we use the monthly surface soil moisture (0-10 cm) from the Noah land-surface model for Phase 2 of the North American Land Data Assimilation System (NLDAS-2) with a spatial resolution of 0.125°× 0.125° to represent the influence of fuel moisture (Xia et al., 2012).

### 2.2.4 Geospatial variables and population

Lastly, population and two geospatial variables are used as predictors, including ecoregions and land cover types which are chosen to capture the effects of land use and ecosystem similarity on wildfire burned area. Land cover mainly describes the physical material at the surface of the earth. The land cover data at the spatial resolution of 30 m is obtained from the 2011 Landsat-derived land cover map from the National Land Cover Database (NLCD) (https://www.mrlc.gov) (Homer et al., 2020). The ecoregion data is obtained from the United States Environmental Protection Agency (US EPA) (https://www.epa.gov/eco-research/ecoregions) (Omernik, 1995; Omernik and Griffith, 2014). The ecoregions denote areas of similarity in the mosaic of biotic, abiotic, terrestrial, and aquatic ecosystem components. Population density data in the year 2010 from the U.S. Census Bureau (https://www.census.gov/geo/maps-data/data/tiger.html) (U.S. Census Bureau, 2010) is used to estimate the influence of present-day human management practices and human activities on wildfires.

## 3. Model

**3.1 Model description**

One major challenge in wildfire prediction is the highly uneven distribution of burned area where the number of grids with large burned areas is typically much smaller than the number of grids with small or zero burned areas (Figure S2a). For the study region (red box in Figure 1), grids without any fire occurrence in combination with those of only small fires (< 25 ha) take up 79% of the total number of the grids but correspond to only 1% of the total burned area. By contrast, grids with

195 the large burned area (>150 ha) account for 84% of the total burned area but only 6% of the total number of grids. For such unevenly-distributed data, standard machine learning methods usually favor the majority class (i.e. non-burned or small fires), leading to the low prediction accuracy of the minority class (i.e. large fires) (Krawczyk, 2016). To alleviate the low bias toward large fires, we develop a model consisting of multiple steps that address the uneven data issue.

Figure 2 demonstrates the structures and processes of our model, which has four steps and uses three machine learning

algorithms. First, for each data grid, given the predictor variables, we use the quantile regression forest (QRF) to predict a distribution of burned area at the targeted percentiles which are chosen at 45, 55, 65, 85, 95, and 99 in this step. The percentiles here refer to the relative position of the predicted burned area in the cumulative distribution of all the burned area data and they are chosen to include the whole conditional distribution. Second, for all the grids, we predict if a grid burns or not by using the logistic regression model and the same set of predictor variables as in the first step. Third, for the grids that are

predicted to burn, instead of predicting burned area directly, we use a random forest (RF) model to predict the percentile of burned area relative to the training set. After all the predicted-burn grids obtain their predicted percentiles of burned area by the RF, the test dataset is divided into six sub-groups according to their predicted percentiles: {(39,49), (50,59), (60,69), (70,79), (80, 89), (>=90)}. The percentile groups are chosen to align with the six percentiles in the first step. The first three percentiles correspond to the median of the first three percentile groups. For example, the first percentile group (39, 49) has a

median percentile of 45, the first percentile of predicted wildfire burned area from the first step. The last three percentiles (85, 95, and 99) from the first step correspond to the last three percentile groups of (70, 79), (80, 89), and (>=90), respectively, although they lie outside the upper bounds of corresponding subgroups. This is based on the assumption that grids with the larger predicted burned area (predicted percentile > 70) in the testing set will have more right-shifted burned area distributions than the distributions of the whole training set, as shown in Figure S3. In step 4, for the grids in a given subgroup, they are

assigned to the burned area value at the corresponding percentiles as determined by the predicted distribution generated from the first step. Specifics of the machine learning algorithms and technical details of the prediction model are described in the subsections below.

Our approach alleviates the issue of unevenness data for two reasons. First, the majority of zero-burn grids are separated by the second step. Second, for the grids predicted to burn, we predict the relative position (i.e. percentiles) of the

220 burned area based on the training set. As Figure S2 and Table S2 show, the distribution of percentiles is less skewed compared to the burned area distribution. Thus, the unevenness of the burned area is less severe when predicting the percentiles than

predicting the burned area directly. Given the possible collinearity between the predictor variables, we choose the logistic model and RF model which are shown to work reasonably well under moderate collinearity (correlation coefficient < |0.7|) (Dormann et al., 2013). We verify that the correlation between any pairs of the time-varying predictor variables is less than 0.7, except for the variables of the antecedent SPEI. We choose to keep the antecedent SPEI covering the different ranges of months to represent the different pre-fire drought conditions which are expected to play an important role for wildfires. For the winter-spring fire season, the pre-fire season starts in October and can range from 3 to 6 months for the start (January) and end (April) of the fire season, respectively. For the summer fire season, we use May as the start month of the pre-fire season and the pre-fire season ranges from 1 to 4 months for the start (July) and the end (September) of the summer fire season, respectively.

### 3.1.1 Random forest regression

Random forest (RF) is an ensemble-learning algorithm built on decision trees. Each tree is built using the best split for each node among a subset of predictors randomly selected at the node (Liaw and Wiener, 2002). The best split criterion is based on selecting the variables at the nodes with lowest Gini Index (GI), which is defined as GI $(t_x(x_i))$ = 1- $\sum_{j=1}^{m} f(t_x(x_i), j)^2$, where $f(t_x(x_i), j)$ is the proportion of samples with the value $xi$ belonging to leave $j$ as node $t$. Two parameters can be adjusted to optimize the RF model, including the number of trees grown ($n_{tree}$) and the number of predictors sampled for splitting at each node ($m_{try}$). The RF regression model first draws $n_{tree}$ bootstrap samples from the original dataset. For each sample, at each node of a tree, $m_{try}$ predictors are randomly chosen from all the predictors and then the best split from among the predictors is determined at each node according to GI. In this study, we have $n_{tree}$ of 1200 and $m_{try}$ of 8 for the winter-spring fire season and $n_{tree}$ of 1500 and $m_{try}$ of 7 for the summer fire season. As the length and characteristics of the two fire seasons are different, we use two sets of parameter configurations for the models of the two fire seasons which include different predictor variables (section 2.2). This way would ensure the prediction model is fully optimized for each fire season to obtain the best prediction accuracy. The predicted value of an observation is the average of the observed values belonging to the leaves of $n_{tree}$ trees. Here, we use the RF model to predict percentiles of burned area for the grids that are predicted to burn.

The benefit of applying the RF model is that it can provide the variable importance that measures the strength of individual predictors. The variable importance is measured by the increase in the mean square error (%IncMSE) and the increase in node purities (IncNodePurity). The %IncMSE is calculated by comparing the mean square error with and without permuting variables for each tree, and the variables with greater values of %IncMSE are more important. As for the IncNodePurity, the changes of residual sum of square (RSS) before and after the split are first derived at each split, and the final IncNodePurity of a variable is obtained by summing over the RSS of all the splits that include the variable over all trees. Thus, a larger IncNodePurity represents higher variable importance.

### 3.1.2 Quantile regression forests

Quantile regression forests (QRF) are an extension of the RF (Meinshausen, 2006). QRF develops trees in the same way as RF, but instead of calculating the average of the values from leaves of the trees to obtain a single predicted value, the QRF estimates the conditional distribution of a target variable. The conditional distribution is calculated by averaging the conditional distributions from all the trees and the predicted quantiles or percentiles are derived from the final empirical distribution function. Here we choose to predict percentiles at 45, 55, 65, 75, 85, 95, and 99 as described above. These percentiles are selected because they can represent the full spectrum of fire sizes ranging from small to extremely large ones. The percentiles less than 45 are typically zero-burn, so the percentile of 45 is the lowest percentile that can possibly record both zero-burn and very small burned area for each grid.

### 3.1.3 Logistic regression model

Logistic regression is used to estimate the probability of wildfire occurrences in a grid cell by the statistical relationships between wildfire occurrences and the predictor variables. Logistic regression is defined as $P_i = \frac{1}{1+e^{-\eta_i}}$ and $\eta_i = \beta_0 + \beta_1 X_{i1} + \beta_2 X_{i2} + \cdots + \beta_\rho X_{i\rho}$, where $Pi$ represents the probability of an occurrence of wildfire in a grid cell $i$, $\eta_i$ is the linear combination of the predictor variables weighted by their regression coefficients ($\beta$), $x_{ij}$ is the value of the predictor variable $j$ of the grid $i$, and $\beta_0$ is the constant. The logit function can be expressed as $\log\left(\frac{P}{1-P}\right) = x_i^T \beta$, where $x_i^T$ is the vector of the predictor variables and $\beta$ is the vector of the parameters. Values of $P$ greater than 0.4 are considered to be an occurrence of wildfires and those equal to or less than 0.4 are interpreted as nonoccurrence of wildfires. If a grid is classified not to burn, the predicted burned area is zero and that grid will not be processed further. On the other hand, if a grid is classified to burn, it would be analyzed by the RF model to predict the burned area percentiles.

### 3.2 Validation method

We apply 10-fold cross-validation (CV) technique to evaluate the model performance and to avoid overfitting. The entire dataset (2002-2015) is randomly divided into 10 equal-sized splits. For each round of CV, the model is trained with nine splits of the data and the trained model is then used to predict burned area at the remaining split.

Classification of burned or unburned grids is evaluated by the accuracy, precision, recall, and F1-score. Precision and Recall are defined in Equation (1) and (2):

$$Precision = \frac{True\ positive}{True\ positive + False\ positive}, \tag{1}$$

$$Recall = \frac{True\ positive}{True\ positive + False\ negaitve}, \tag{2}$$

where true positive is the number of burned grids correctly predicted, false positive is the number of grids which are unburned but are predicted as burned, and false negative is the number of grids that are burned but are predicted not to burn. The F1 score measures a model's accuracy that combines precision and recall:

$$F1 = \frac{2}{recall^{-1} + precision^{-1}},\qquad (3)$$

F1 score has a maximum value of 1 and a minimum value of 0, and the higher F1 indicates a higher balance between Precision and Recall. In addition to the aforementioned evaluation criteria, we use the receiver operating characteristic (ROC) curve, and the area under the curve (AUC) statistics to evaluate the classifier (Metz, 1978). The ROC curve shows how well the model can distinguish between the true positive rate (TPR) and the false positive rate (FPR), where TPR and FPR are expressed by Equation (4) and (5):

$$True\ positive\ rate = \frac{True\ positive}{True\ positive + False\ negative},\qquad (4)$$

$$False\ positive\ rate = \frac{False\ positive}{False\ positive + True\ negative},\qquad (5)$$

The AUC is the area under the ROC curve and it ranges from 0 to 1. The greater the AUC, the better discrimination between true positive and true negative.

Burned area predictions are evaluated using statistical indicators such as the coefficient of determination ($R^2$), mean absolute error (MAE), and root mean squared error (RMSE) between the predicted and observed wildfire burned areas. The evaluation is conducted for the winter-spring fire season and summer fire season separately. The prediction performance is also quantified in terms of the model ability in reproducing temporal variation of burned area for each grid and spatial patterns of burned area across all the grids of the study domain. Details on the calculation of the spatial and temporal correlations are described in the Supporting Information.

## 4. Model validation and evaluation

Here we present the validation results at two spatial scales: the grid-scale of 0.5°× 0.5° and the large-domain scale of 700 km x 700 km corresponding to the size of the study domain (red box in Figure 1). The grid-scale prediction of all possible outcomes (i.e., unburned, small burned, and large burned area) is a unique strength of our model. To the best of our knowledge, only few previously published studies included unburned and small burned grids into the prediction of wildfire burned area at a grid-scale as fine as 0.5°× 0.5°. At the large-domain scale, we will compare our model performance with prior studies that predicted total burned area of an ecoregion or a country.

Table 2 lists a variety of statistics representing the model performance at the grid-scale for the winter-spring fire season and summer fire season. The prediction performance of the classifier (i.e. the second step in the model) is evaluated by the ROC curves (Figure S4), the area under the ROC curve (AUC), accuracy, recall, precision, and F1-score. The ROC curves of both fire seasons steer toward the upper left corner, indicating good performance of the model with a high detection rate of

fires and a low false alarm. The AUCs for the two fire seasons are 0.82 and 0.83. The accuracy and F-1 score are 0.74 and 0.79, respectively for the winter-spring fire season and 0.74 and 0.77 for the summer fire season. These results indicate the model is capable of classifying burned grids and unburned grids with a good balance of recall and precision.

In terms of burned area prediction at the grid-scale, the $R^2$ reaches 0.42 and 0.40 for the winter-spring and summer fire season respectively. MAE and RMSE are 1.13 and 8.37 respectively for the winter-spring fire season, and 0.57 and 4.26 for the summer fire season. Before comparing these prediction statistics with previously published studies that predicted gridded burned area, it is important to note that the prediction accuracy will depend on the temporal scale (e.g. monthly or annual) and grid resolution at which the prediction is made. The larger spatiotemporal scales are expected to have a better prediction performance. Regarding the type of grids to be predicted, the most challenging case is the prediction including all possible outcomes of a given grid (i.e., unburned, with small burned areas, and with large burned areas). As fewer prior studies of the similar nature as ours predicted all possible outcomes (i.e. not only large burned areas but also unburned and small burned cases) at the grid-level and none of these studies targeted the South Central US, we choose to compare our model performance with previously published models that predicted gridded burned area in terms of the approaches, the temporal and spatial resolution, and the percent of variance explained by the model, regardless of their study regions, periods, methods, and predictors. Chen et al. (2016) used ocean climate indices to estimate annual burned area at the grid resolution of 1° x 1° but their prediction was only for those grids with non-zero annual burned area. They achieved a prediction $R^2$ of less than 0.3 (correlation coefficient r around 0.55) over the southern US (SUS). Using boosted regression trees, Liu and Wimberly (2015) obtained a higher $R^2$ of 0.76 between climate variables and burned area over the western US, but their investigation was limited to only extremely large fires (> 405 ha) and was at a 1° x 1° resolution and annual timestep. Compared to those studies, our model targets a more challenging prediction (i.e. prediction at a finer spatial and temporal scale and for all the grids), yet achieves a comparable if not better performance at the grid scale.

Considering there are very few studies that predicted burned area by grids and at the same time considered unburned grids or grids with small fires, we extend the comparison to past studies predicting burned area of regions with the similar spatial scales of 0.5° x 0.5°. Urbieta et al. (2015) used Multiple Linear Regression (MLR) to predict the annual burned area of provinces and national forests in the southern countries of the European Union (EUMED) and Pacific Western US (PWUSA), with the mean domain size of 108 km x 108 km. Their reported median $R^2$ is 0.28 for EUMED and 0.22 for PWUSA, smaller than our value (0.4). Using the MLR method, Carvalho et al. (2008) predicted monthly burned area of Portuguese districts of sizes ranging from ~ 25 km x 25 km to 100 km x 100 km and their $R^2$ is between 0.43 to 0.80. The better model performance was only for some districts with evenly-distributed burned area, whereas the districts with highly right-skewed burned area distributions (Evora and Portalegre) had prediction $R^2$ of 0.43~0.45. Bedia et al. (2014) predicted monthly burned area of the phytoclimatic zones in Spain (~25 km x 25 km to 100 km x 100 km) by using multivariate adaptive regression splines (MARS) and obtained $R^2$ ranging from 0.01 to 0.37. In comparison with these results, the $R^2$ of 0.42 and 0.40 that we achieve for the two fire seasons at a grid resolution of 0.5° x 0.5° is a significant improvement for situations with unevenly-distributed burned

area. In addition, by predicting all possible outcomes for all the grids within a large domain, our model framework would be more flexible and practical to be applied to other domains.

The aforementioned statistics demonstrate the general capability of our four-step model in predicting gridded burned area over the study period. We select three specific years to further illustrate the model performance: 2011 with the largest domain-mean gridded burned area, 2008 and 2014 with the domain-mean gridded burned area close to the 14-year-mean for the winter-spring and summer fire season respectively (Table S4). Figure 3 shows the selected CV-predicted and observed monthly burned area of these years for each fire season. The $R^2$ is 0.42, 0.51, and 0.66 for 2011 (combing both seasons), 2014

(the winter-spring season), and 2008 (the summer fire season), respectively, after excluding misclassified grids. MAE of 2011, 2014, and 2008 are 5.25, 0.77, 0.43 and RMSE are 21.06, 5.87, and 1.75. The detailed statistics of the model performance for each year are also shown in Table S5. The results show that the model has a better performance in predicting gridded burned area for normal years of 2008 and 2014 than for the exceptionally large wildfire year of 2011. Although larger MAE and RMSE are shown in 2011 (peak year), our model predicts significantly larger mean gridded burned area for the peak months.

For 2011, the large burned area can be well modeled but the small burned area (log of burned area < 2) is overpredicted. This can be explained by the fact that the extremely hot and dry weather during 2011 caused fire-favorable conditions across the study domain. Due to the lack of reliable and detailed information about ignition and suppression, it is difficult for the model to discriminate between small and large fires given widespread extreme drought conditions across the whole domain during 2011 (Long et al., 2013; Nielsen-Gammon, 2012).

The model performance is further evaluated in terms of its ability in reproducing the spatiotemporal patterns of monthly mean burned area for the two fire seasons (Figure 4). The correlation coefficient between the 14-year mean observed and predicted burned area is 0.82 and 0.80 for the winter-spring and summer fire season, respectively. For the whole study period, more than 60% of the months have a spatial correlation larger than 0.5 for both fire seasons between the observed and predicted monthly burned area. It is noteworthy that such performance is achieved without introducing any coordinate variables like longitude or latitude as predictors. This indicates the chosen predictors contain sufficient information to capture the spatial

heterogeneity of the environmental factors and thus the framework of the model could be easily adopted for other regions, making it possible to be incorporated into climate models in future applications. Temporally, more than 70% of the grids have a correlation higher than 0.5 between the observed and predicted time series of burned area (combined the two fire seasons) (Figure S5). These results demonstrate the model has a certain ability in predicting both spatial and temporal variation of the

burned area at the grid-scale across the study domain.

        Even though bias may be introduced in the multi-steps model, the developed four-step model can achieve higher accuracy and alleviate the issue of uneven-distributed dataset. To prove that, we compare the model performance of our four-step model with the prediction performance of simulations using MLR, only the RF model and another decision-tree-based ensemble machine learning algorithm called eXtreme Gradient Boosting (XGBoost) (Chen and Guestrin, 2016). The results

are listed in Table S2 and the description as well as the parameters of XGBoost are included in supplementary (Table S3). Our four-step model has a lower MAE, which is 27% and 33% lower than the MLR model for the winter-spring and summer fire

season, respectively. Compared to the RF model, our four-step model has a lower MAE by 15% and 19% for the winter-spring and summer fire season, respectively. Compared to the XGBoost model, the MAE from our four-step model is 11% and 15% lower for the two fire seasons. The distribution of MAE from the 10-fold cross-validation shows that our four-step model has a smaller median MAE but a larger range of MAE compared to other models (Figure S6). In addition, the distribution of percentiles is more uniform than the distribution of the burned area, as shown in Figure S2 and the skewness value. Details about the calculation of skewness are described in Supporting Information. Larger positive skewness value indicates a more highly right-skewed distribution. The skewness of the burned area is 37.4 and 33.8 for the winter-spring and summer fire season while the skewness of percentiles is 0.7 and 0.96, showing that the strategy of the four-step model can effectively reduce unevenness of the distribution.

In addition to the grid-scale statistics, we evaluate the model performance at the large-domain scale by adding up all the grid-level predictions to obtain the total burned area of the study domain by months. Figure 5 shows the time series of the predicted total burned area over South Central US in comparison to the observed ones for the two fire seasons. The domain-scale prediction explains 50% and 79% of the month-to-month variability of burned area for the winter-spring and summer fire season, respectively. Higher $R^2$ for the summer fire season can be explained by the stricter fire regulations during summer in the southern states, such as Texas (While and Hanselka, 2000). For the summer fire season, under strict fire regulations, environmental factors such as high temperature or low relative humidity can play a more important role in wildfire development. For the winter-spring fire season, more human perturbations may be involved. As the human factor in the model does not capture such perturbation, less variability is explained by the model for the winter-spring season. MAE of the monthly burned area across the whole domain is 251.3 km$^2$ for the winter-spring fire season and 100.7 km$^2$ for the summer fire season. Generally, our model is able to capture the interannual variability of burned area and the prediction accuracy of our model in terms of $R^2$ is equivalent to or better than most of the published studies on the ecoregion scale or country scale, as shown in Table S1.

## 5. Contributions of environmental factors to predicted wildfire burned area

### 5.1 Individual variable importance at grid scale

Before discussing the environmental controls on wildfire burned area across the study domain, it is useful to understand the dominant factors controlling the burned area at the grid scale. One advantage of the random forest approach is that it provides the variable importance metrics that can measure the power of predictor variables in the prediction. Figure 6 shows the top 14 predictors ranked by %IncMSE to illustrate the intricate relationships between fires, weather, climate, and fuel. The top 14 variables are chosen because they represent the top quarter (25%) of the selected predictor variables. In addition, a sensitivity test shows that the largest drop in the %IncMSE occurs around the 15th variable ranked by importance, as shown in Table S6. To ensure the reliability of the inferred importance of predicted factors, we conduct 50 times 10-fold

cross-validation by randomizing the order of all the data each time. Figure S7 shows the distributions of %IncMSE for each variable ranked by the median %IncMSE. Even though the numerical values of feature importance vary in different runs, the variable ranks by median values stay the same, indicating the robustness of the feature importance identified by the RF model.

For both fire seasons, RH anomaly is the most important predictor of wildfire burned area at the grid-scale (Figure 6). This finding broadly supports past studies that highlighted the importance of RH on burned area (Riley et al., 2013; Ruthrof et al., 2016). Yet, our model particularly reveals the response of fire burned area to the changes in RH anomaly, which is a climate variable as opposed to a weather variable. The rhum is the actual RH which can vary by location and season, while RH anomaly measures the departure of rhum from its long-term average due to climate change and/or climate variability. For the study domain and time period, the correlation between RH anomaly and RH is 0.66. Although they have a moderate correlation, their values have different physical meanings and both of them are included in the model. For example, for grids with rhum of ~70%, rhum_anomaly can range from -11.16% to 15.35%. For the same rhum value of ~70%, positive rhum_anomaly indicates a relatively wetter condition and negative rhum_anomaly a relatively drier condition compared to their long-term condition in the past. The variable importance metric highlights that RH anomaly, which indicates the changes of the fire-season RH relative to its historical climatology, ranks higher than the actual value of the fire-season RH.

While both fire seasons have RH as the top driver of burned area, notable differences are found for the relative importance of other variables between the two fire seasons. For the summer fire season, temperature anomaly and maximum temperature anomaly are the other two climatic factors besides RH anomaly that are included in the top 14 variables. While RH anomaly and temperature anomaly are expected to correlate to some extent, the slope from a linear regression of RH anomaly (y) on temperature anomaly (x) is substantially greater (in absolute value) in the summer fire season (slope= -3.7) than that in the spring fire season (slope= -0.89) (Figure S8). This highlights the stronger dependence of RH anomaly on temperature anomaly in the summer. Additionally, larger burned areas (75[th] percentile and above, black dots in Figure S8) mainly occur under the condition of low RH anomaly and high temperature anomaly (bottom-right corner), in particular for the summer fire season. The results suggest that higher temperature coupled with lower relative humidity can cause drier fuel and create favorable conditions for fires to start, spread, and burn more intensely, in particular during the summer fire season (Williams et al., 2013; Holden et al., 2018).

For the winter-spring fire season specifically, the long-term averages of monthly total precipitation and monthly means of daily precipitation (apcp_avg and asum avg) are identified as the key climate variables (Figure 6a). These two variables represent the precipitation normal, indicating the amount of available moisture that could affect fuel distributions and tendency of fire activities (Keyser and Westerling, 2017; Westerling and Bryant, 2008). The averaged SPEI of the preceding 4 months is the second most important variable and the highest-ranked weather variables, which is even more important than the SPEI during the fire season. The averaged SPEI of the preceding 3 months and 5 months are also included in the top 14 variables. The 3-5 months' time lag coincidentally corresponds to the interval between the two fire seasons. Thus, our results indicate that burned area in this season is highly dependent on the pre-fire-season drought conditions, which is in agreement with prior studies (Scott and Burgan., 2005; Riley et al., 2013; Turco et al., 2017). To better understand how the

changes of top variables affect the burned area, we use the partial dependence plots to show the marginal effect of a variable on the prediction performance of the built model (Friedman, 2001). Figure S9 shows the partial dependence plots of the top

four variables (RH anomaly, SPEI_mean4m, apcp_avg, and temp_sd) for the winter-spring fire season. For RH anomaly, the fitted logarithmic burned area becomes larger if the RH anomaly is smaller than 2% (Figure S9a). This change likely indicates the sensitivity of burned area to the fire-season moisture. The similar pattern is also shown in the partial dependence plot of the mean SPEI of the preceding 4 months (Figure S9b). Larger fitted burned area is observed to be associated with the preceding SPEI smaller than zero, suggesting that burned area in this season is highly dependent on the pre-fire-season drought

conditions. As for the average precipitation of 1979-2000, the fitted burned area increases as the average precipitation increases (Figure S9c). This implies the shift of fire regimes in that larger fires occur in the areas with more average precipitation in the past. For the standard deviation of temperature during 1979-2000, the fitted burned area declines dramatically when the standard deviation of temperature is larger 9K, suggesting the threshold effect of temperature variation on the burned area in the winter-spring fire season (Figure S9d). In addition to the top 4 variables which are all meteorological variables, the average

of LAI and sum of neighboring LAI for months $t-1$ to $t-6$ are the only fuel variables that are selected among the top 14 variables in the winter-spring fire season (Figure 6). Although these two variables rank below others among the top 14 variables, they are the fifth and sixth most important variables when excluding the fixed variables. Thus, when considering the importance of the time-varying variables, we can infer that fuel abundance together with drought conditions in the pre-fire-season determine the amount of dry fuel, which likely exerts the primary controls of the burned area during the winter-spring fire season.

For the summer fire season, important weather variables include the average of monthly accumulated precipitation of the preceding one month and the mean SPEI of the preceding one month, two months, and three months (Figure 6b). These variables are known to affect burned area by influencing fuel moisture. Consistently, fuel moisture as represented by soil moisture is identified as the only fuel variable among the top 14 variables in the summer fire season. These results suggest that fuel drying during the summer fire season driven by both increasing temperature and pre-fire season drought conditions is the

pivotal process determining wildfire burned area in the summer. Similar to our findings, rising summer temperature under climate change was found to cause fast fuel dryness and increase fire activity in the western US (Williams et al., 2013; Holden et al., 2018). As the partial dependence plots show (Figure S10), the large burned area is associated with low values of RH anomaly, minimum RH anomaly, the mean SPEI of the preceding 2 months, and long-term (1979–2000) standard deviation of temperature for the summer fire season. The fitted logarithmic burned area increases rapidly as the RH anomaly decreases

toward zero and the increase in burned area reaches a maximum at RH anomaly of –14% (Fig. S10a). Compared to the partial dependence plot for RH anomaly, the fitted burned area increases more rapidly with decreasing minimum RH anomaly (Fig. S10c). At below zero, the sensitivity of log(burned area) to the minimum RH anomaly is 0.04 %$^{-1}$ (Fig. S10c), while the corresponding sensitivity to RH anomaly is only 0.02 %$^{-1}$ (Fig. S10a). The stronger sensitivity of burned area to minimum RH anomaly indicates the stronger effect of extremely low humidity conditions on fire growth as compared with the mean RH

conditions. For the standard deviation of temperature during 1979-2000, larger burned area is observed with smaller standard deviation of temperature in the past. This suggests burned area would become larger for the grids with less variation of

temperature in the summer. As for the mean SPEI of the preceding 2 months, we see an increase of fitted burned area at zero, with the largest increase at –1.8, which supports the importance of fuel drying process in the summer fire season. For both fire seasons, RH anomaly, mean SPEI of preceding months, and standard deviation of temperature for 1979-2000 are selected as the top 4 predictors, highlighting the common importance of these variables in the two seasons but with different thresholds and magnitudes in their effects on burned area. The difference in controlling factors for wildfires between the two fire seasons can be also demonstrated by the difference in correlation coefficients between burned area and predictors in the two seasons. The correlation between burned area and the average daily precipitation of months $t-1$ is -0.05 and -0.28 for the winter-spring and summer fire season respectively. The correlation between burned area and the average of SPEI of pre-fire seasons (months of $t-1$ to $t-3$ for winter-spring and $t-1$ to $t-2$ for summer) is -0.28 and -0.34. Although lower moisture during the pre-fire season increases burned area for both fire seasons, the summer fire season has a stronger negative correlation between burned area and moisture during the pre-fire season. For the summer, since vegetation is relatively sufficient, fuel drying in the fire season and pre-fire-season is a more important control for wildfire development. For the winter-spring fire season, as the vegetation amount is not as abundant as in the summer fire season, both fuel abundance and fuel drying in the pre-fire-season are critical for wildfires development. The balance between the two factors may explain the weaker negative correlation between burned area and moisture in the pre-fire season for the winter-spring fire season.

Figure S11 shows the correlation coefficients between the predictor variables. Most of the important variables have weak to moderate correlations ($r < |0.7|$) between each other. The exceptions are for the fixed-climate variables (e.g. asum_avg vs. apcp_avg and temp_sd vs. tmax_sd) and the antecedent variables (e.g. SPEI_mean4m and SPEI_mean5m) for both fire seasons. This is expected because the long-term mean or standard deviation of the same types of meteorology do not change by time and the average of antecedent drought conditions (SPEI) may not vary a lot from including or excluding a single month. Although there is collinearity between the predictor variables, the logistic model and the RF model we use in this study are relatively insensitive to collinearity. Random forest as a machine learning tool is less unaffected by the issue of multicollinearity than traditional regression methods because the random forest model randomly selects predictors used for each tree so that the probability of sampling strongly correlated variables in a particular tree is largely avoided (Siroky, 2009). To prove that the collinearity would not be an issue for our model, we calculate Variance Inflation Factor (VIF) for the random forest model by a bootstrapping of seven predictors (the number of predictors used in each tree) out of all 58 potential predictors for 5000 times. Each sampling yields seven VIF values, and hence we can obtain a distribution of 35000 VIFs for 5000 samplings. Figure S12 shows the distribution of VIFs for all the predictors. The distribution has a median of 1.67 for the winter-spring and a median of 1.62 for the summer fire season. The distribution has about 96% of the VIF values smaller than 10 for both seasons, demonstrating the minimized multicollinearity in the random forest model. In addition, we conduct a sensitivity test where the model uses predictor variables that have lower degrees of collinearity ($|r|<0.5$), compared to the results using variables with higher degrees of collinearity ($|r|<0.7$). The results show that removing the predictors that have a higher degree of collinearity causes larger biases in the classification of burned grids and the prediction of extremely-large fires (Table S7). The overall MAE and RMSE are also slightly degraded in the sensitivity test. That is because although some

variables may have a moderate correlation, they have different physical meanings and thus provide different predictive information. Therefore, we include all the variables in the model and allow the algorithms to choose the predictors for better performance.

Overall, the analysis of variable importance and partial dependence plots reveal the common and different characteristics of the wildfire development between the two fire seasons and show semi-quantitatively that drought conditions in the preceding months (3-5 months for the spring fire season and 1-3 months for the summer fire season) may be more important than within-season conditions. Furthermore, we demonstrate that the effect of climate variability on burned area is consequential and even more influential than concurrent fire weather. This aspect has not been well documented or quantified in past studies for the South Central US, partly due to a lack of long-term observations of wildfires over this region. Although we did not use long-term wildfire data (only 14-years of data used), with the 10-fold cross-validation approach, the training dataset contains around 16277 samples for each fold. Such a large sample size is enough to capture the variability in wildfire activity and its response to the recent decadal climate if we assume wildfire relationships with the environmental factors contain certain uniqueness for each individual grid. Considering the majority of grids over the study domain are grassland/plain with short fire interval (~1 year) (Barrett et al., 2010), the 14-year data is suitable for assessing fire variability for our study domain. Within this 14-year period, some regions (e.g. SE Texas) experienced the largest wildfire and the most severe single-year drought in the past 50 years (i.e., 2011 Texas wildfire). For future applications, our model can be applied to other regions with longer fire return intervals if more data is included. As the accuracy of our model is not quite high, uncertainties may exist in the rank of variable importance from the RF model. However, the selected top 14 variables all have physical linkages to wildfire burned area and they have been discussed in this section and prior studies.

## 5.2 Relative importance of environmental controls at large scale

The variable importance metrics presented in the previous section reveal the relative importance of individual predictors. As mentioned before, these predictors are purposely selected from four broadly defined categories of environmental controls on wildfire burned area, namely climate, weather, fuel, and fixed-geospatial. Here the climate category includes only variables of climate anomalies. The weather and fuel category are comprised of both fire-season and antecedent weather and fuel conditions, respectively. The fixed geospatial category includes all the variables that do not change with time, including land types, ecoregion types, population, and 22-year means and standard deviations of meteorological variables (i.e. climate normals). Given that variables within the same category may work in conjunction to create conditions conducive to wildfires, in this section we examine the composite influence of predictors by category and quantify the contributions of these environmental controls to wildfire burned area. To do so, the prediction model developed from Section 3 is used to decompose the effect of different environmental controls across our study domain by perturbing all the variables belonging to one category at a time. The details of the decomposition method are described in the supplementary information.

Figure S13 shows the time series of the contributions of different environmental controls on the burned area for the two fire seasons. The results show that the weather, fuel, climate, and fixed effects tend to increase the burned area for the

large burn events (e.g. July 2011 in the summer fire season). To further investigate whether or not all factors would increase the burned area, we calculate the effect of each group in percentage by dividing the total burned area of the month, as shown in Figure S14. For the months with the large burned area (e.g. Jan 2006 and Sep 2011), weather, fuel, climate, and fix effect tend to increase burned area. This is consistent with the results in Fig S8. This is not the case for some months with the relatively small burned area, such as Feb 2012 where the interaction (-143%), climate (-1.4%), and weather effect (-33.8%) reduce the burned area but fuel (12%) and fix effect (266%) together increase the burned area. As the number of variables in each environmental control category is different, we first normalize the absolute contribution of one environmental control by the number of variables in that category and then compare each category's contribution in scaled absolute percentage, which is defined as the normalized absolute contribution of one environmental control divided by the summation of normalized absolute contributions over all the categories. The scaled absolute percentage represents the average contribution from all the variables in one environmental category, so the variable importance presented here is not affected by the number of variables we include in each category. Figure S15 shows the time series of the scaled absolute percentage of each category. For both fire seasons, on average, the climate and fixed categories have larger contributions to the burned area than other categories, although their relative importance varies by time. Figure 7 and Table S8 present the mean effect of the environmental controls where the scaled absolute percentage of each category of environmental controls is averaged over the whole study periods. Figure 7 clearly shows that the climate category on average has the largest contribution to the burned area for both fire seasons, with the mean scaled absolute contribution of 33% and 35% for the winter-spring and summer fire season, respectively. This suggests climate variability is a significant factor to explain wildfire burned area over our study domain. This result is consistent with previous studies that demonstrated the significant contribution of changing climate to the total burned area of ecoregions in the western US (Littell et al., 2009; Swetnam and Anderson, 2008; Yue et al., 2013). For example, increasing temperature and earlier spring snowmelt due to climate change are highly associated with increased large wildfire activity in the western US (Westerling et al., 2006). Another study showed that fire-year climate variables such as average spring temperature are predictive variables that could improve the predicting probability of high severity fires in the western US (Keyser and Westerling, 2017). Additionally, the fixed effect that comprises the geospatial variables and past climatology is ranked as the second most important control (Figure 7). This is consistent with the findings of Keyser and Westerling (2017), which revealed the importance of long-term climate normals in controlling large fire occurrences in the western US.

Comparing the effects of the environmental controls between the two fire seasons, we find the fuel effect is significantly more important in the winter-spring fire season, while weather and climate effects are more substantial in the summer fire season. This can probably be explained by the different characteristics of the two fire seasons. As biomass growth is relatively limited in the winter-spring fire season, the effect of fuel (mainly from vegetation in the pre-fire growing season) is likely the limiting factor for wildfires. On the other hand, vegetation is relatively sufficient during the summer growing fire season and thus fuel abundance would not be a constraint of wildfires (Littell et al., 2009; Zhang et al., 2014). Yet, fire weather that determines fuel moisture is a substantial factor in the summer fire season (Figure 7).

The above analysis represents the relative importance of the environmental controls at the large-domain scale. At the grid scale, we calculate the average of variable importance (%IncMSE) from RF (section 3.1.1) of each category and use the category-averaged variable importance to represent the relative importance at the grid-scale (Table S9). Climate variables are found to have the largest importance in controlling burned area at the grid scale for the two fire seasons, with the mean %IncMSE of 12.09 and 19.18 for the winter-spring and summer fire season, respectively. This is consistent with the results based on the large-domain scale. Fuel effect outweighs weather effect on the grid scale in the winter-spring fire season, while weather effect is more important in the summer fire season, both consistent with the aforementioned analysis based on the large-scale domain (Table S9). However, the fixed effect estimated at the grid-scale is less important than at the large-scale domain (Table S9) and this is partly due to how these variables are encoded in the model. Fixed variables consist of past climatology and geospatial variables (i.e. land use, ecoregion, and population). The geospatial variables, except population, are encoded as categorical variables in the prediction model. For example, forest ecoregion is coded as 0 or 1 for a given grid, with 0 representing non-forest and 1 representing a forest. For such an encoding method, each categorical variable (e.g. forest v.s. non-forest) tends to have a smaller relative importance score, compared to the relative importance score of other variables encoded by continuous values. As RF measures the effect of a specific split on the improvement in model performance and aggregates the improvement of all the splits with a specific variable, the fragmented scores for each category are likely smaller than the scores reflecting all of the categories. Therefore, for the relative importance at the grid level measured by RF, the effect of a single geospatial variable such as a land type on the burned area is trivial. When we average the relative importance of all the fixed variables including many small scores, the resulting average importance becomes still a small value.

## 6. Concluding remarks

We present a model consisting of multiple machine learning methods to predict monthly burned area over South Central US at 0.5° x 0.5° grid cells. The prediction model is able to alleviate the issue of unevenly-distributed burned area and consequently improves the model capability of predicting large burned area at a finer spatial and temporal scale. The predicted burned area shows a good agreement with the observed burned area at both the grid and large-domain scale. At the grid scale, the classification component of the model achieves an AUC of 0.82 and 0.83 for the winter-spring and summer fire season, respectively. With respect to burned area prediction, a CV-$R^2$ of 0.42 and 0.40 is achieved for the winter-spring and summer fire season, respectively, which makes a significant improvement to the prediction for the cases with unevenly-distributed burned area compared to most past studies. Our four-step model is able to predict the spatial patterns of the 14-year mean burned area, with a correlation coefficient between mean observed and predicted burned area of 0.82 and 0.80 for the winter-spring and summer fire season, respectively. Throughout the study period, more than 60% of the months have a spatial correlation larger than 0.5. When comparing the timeseries of observed and predicted burned area of each grid across the study domain, over 70% of the grids have a correlation coefficient larger than 0.5. At the large-domain scale, the prediction model

can explain 50% and 79% of the interannual variability of wildfire burned area for the winter-spring and summer fire season,
respectively. The validation results demonstrate that the model has certain skills in predicting monthly burned area at both grid-scale and large-domain scale.

Although the model shows a better ability to predict monthly burned area at both grid-scale and large-domain scale than past studies of similar nature, it has several limitations. First, errors might be propagated through our serial model and lead to lower accuracy. For example, when the burned grids are predicted not to burn, low bias occurs because the burned grids
are not able to enter step 3. Similarly, inclusion of unburned grids in step 3 will introduce a positive bias. Second, random forest or quantile regression forest cannot predict burned area greater than it observes before, i.e. the maximum burned area of any of the available grids. We should point out that such limitation is applicable *only at grid level* and that upper limit is taken from all available grids of the whole training period, which we refer to as the global upper limit per grid. For example, the global upper limit is 514 km$^2$ per grid for the winter-spring fire season, and 238 km$^2$ per grid for the summer fire season. For
a single grid, burned area prediction can be greater than what this grid had experienced before by learning from other grids, although the prediction per grid cannot exceed the global upper limit. Figure S16 shows an example for a randomly selected grid box. For this grid, the model predicts the largest burned area on Feb 2008, consistent with observed burned area. This demonstrates that any single grid can predict burned area larger than the grid maximum by learning from other grids and as such a larger total burned area *for the domain* can be predicted by the model under future climate change. In addition, we
verify that the global upper limit is a sufficiently large value because of the intrinsically skewed nature of burned area distributions. Figure S17 shows the distribution of gridded burned area for year 2011, an extremely severe fire year for the study domain, in comparison to the distribution of all other years during 2002-2015. It can be seen that the majority of the burned areas for the extreme year are still within the range of the observed burned area in 2002-2015, with only two grids having burned areas larger than the global upper limit from 2002-2015 (excluding 2011). The total burned area of those
exceedance grids only accounts for 20% of total burned area for 2011, which is within the stated uncertainty range of our prediction model. Third, as machine learning models are data-driven, data quality of different input datasets may introduce biases as the input datasets come from a wide variety of data sources and errors in one type of input data may cause sequential errors in the prediction. For instance, biases in the NARR meteorological data can further lead to incorrect fire-meteorology relationships learned by the model. Fourth, this study focuses on the effects of environmental controls on burned area under
present-day human management practices and human activity. As such, we do not examine the effects of time-varying socioeconomic factors on burned area, such as human actions that affect wildfires through ignition, suppression, or modifying fuel distribution (Andela et al., 2017; Bowman et al., 2011; Mann et al., 2016; Syphard et al., 2007). Given that human activity is one of the major controls on fire activity, future work is needed to better understand the role of human activity engaged with climate change and its implications for wildfire control. Finally, the pre-defined parameters that are used in the model,
including the percentiles and subgroups, may induce uncertainties. To understand the related uncertainties, we switch the pre-defined percentiles but fix the subgroups in the first sensitivity experiment (Table S10). In this experiment, the last three quantiles are changed to the median values between a new set of lower and upper bounds. The second experiment is conducted

by changing the number of subgroups, their ranges, and the corresponding percentiles. Generally, changing pre-defined parameters has little effect on overall MAE for the two fire seasons but the MAE of large burned area becomes larger and the standard deviation of the predicted values becomes smaller. Thus, the pre-defined parameters mostly affect the spread of the predictions and the prediction of large burned areas. Despite this sensitivity, the prediction model with the chosen settings (i.e. percentiles and subgroups) is able to predict burned area at 0.5° x 0.5°-grid scale and achieves a higher prediction accuracy compared to prior studies.

The individual variable importance from the RF model is analyzed and discussed. For both fire seasons, RH anomaly followed by drought conditions in the preceding months (3-5 months for the winter-spring fire season and 1-3 months for the summer fire seasons) are the two top variables in predicting burned area at the grid scale. For the winter-spring fire season specifically, the average of LAI and sum of neighboring LAI of the preceding six months are the only two fuel variables that are identified in the top 14 variables and they rank fifth and sixth when only considering time-varying variables. The findings suggest that fuel abundance together with drought conditions during the pre-fire season regulate the abundance of dry fuel, which is the primary control of fire burned area during the winter-spring seasons. For the summer fire season, temperature anomalies, the average of monthly accumulated precipitation of the preceding one month, and fire season soil moisture are important variables in predicting burned area. This suggests that temperature variability and pre-fire season drought can speed up fuel drying and lead to wildfires in the summer. The model highlights the effect of climate variability on burned area as well as the different environmental controls of burned area for the two fire seasons.

Besides the relative importance of individual predictors, we also analyze the relative importance of the environmental controls by four categories - climate, weather, fuel, and fixed-geospatial - at both the grid and large-domain scale. The relative importance of these factors is generally consistent at the two scales. The climate variable on average has the largest contribution to the burned area for both fire seasons, with the mean scaled absolute contribution of 33% and 35 % to the burned area at the large-domain scale for the winter-spring and summer fire season, respectively. For the winter-spring fire season, the fuel variable on average has larger importance compared to the weather variable; while for the summer fire season, the weather variable is more dominant than the fuel variable. The difference in the relative importance of the environmental controls between the large-domain scale and grid scale mainly lies in the predominance of the fixed effect. The fixed effect is ranked as the second most important control at the large-domain scale, but it is not as important at the grid scale.

Predictor variables representing climate variability are ranked as the most important variables by our prediction model. This reinforces the importance of regional climate variability as the key driver for wildfires that have been revealed by past studies for other regions, yet our study is among the first to explicitly demonstrate such importance for the South Central US. For this region, our model further reveals drought conditions in the preceding 3-5 months of a fire season as an important predictor for wildfire burned area. This antecedent time scale would be valuable for fire management and fire prediction in the future. While the relative importance of environmental controls is largely consistent between the large-domain scale (~700 km x 700 km) and the grid scale (~50 km x 50 km), our analysis at different spatial scales would help estimate how the relationship

between wildfire and environmental controls will change as a function of spatial scales, which could be used to improve wildfire modeling and prediction in different models.

*Code availability.* Model code is available upon request to the first author


*Data availability.* All dataset used in this study are publicly accessible online at https://dataverse.harvard.edu/dataset.xhtml?persistentId=doi%3A10.7910%2FDVN%2FLRPDAA

*Author contributions.* SW and YW conceived the research idea. SW wrote the initial draft of the paper, performed the analyses,
and model development. All authors contributed to the interpretation of the results and the preparation of the manuscript.

*Competing interests.* The authors declare that they have no conflict of interest.

*Acknowledgements.* This work was funded in part with funds from an AI for Earth grant from Microsoft and from the State of
Texas as part of the program of the Texas Air Research Center (grant number: 117UHH0175A). YW acknowledge additional support from NOAA Atmospheric Chemistry and Carbon Cycle Program (NA19OAR4310177). The contents do not necessarily reflect the views and policies of the sponsor nor does the mention of trade names or commercial products constitute endorsement or recommendation for use. We acknowledge the NCEP Reanalysis data provided by the NOAA/OAR/ESRL PSL, Boulder, Colorado, USA, from their website at https://psl.noaa.gov/.

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

**Table 1.** Predictor variables that were used in the fire prediction models

| Variables | Abbreviation | Categories | Temporal resolution | Spatial resolution | Data source |
|---|---|---|---|---|---|
| **Weather variables** | | | | | |
| Monthly mean surface temperature | temp | weather | monthly | 32 km | North American Regional Reanalysis (NARR) |
| Monthly mean of daily precipitation | apcp | weather | monthly | 32 km | North American Regional Reanalysis (NARR) |
| Monthly total precipitation | asum | weather | monthly | 32 km | North American Regional Reanalysis (NARR) |

| | | | | | |
|---|---|---|---|---|---|
| Monthly mean surface relative humidity (%) | rhum | weather | monthly | 32 km | North American Regional Reanalysis (NARR) |
| Monthly mean U-component of wind speed | U | weather | monthly | 32 km | North American Regional Reanalysis (NARR) |
| Monthly mean V-component of wind speed | V | weather | monthly | 32 km | North American Regional Reanalysis (NARR) |
| Monthly maximum temperature | tmax | weather | monthly | 32 km | North American Regional Reanalysis (NARR) |
| Monthly minimum RH | rmin | weather | monthly | 32 km | North American Regional Reanalysis (NARR) |
| Number of consecutive days without rainfall in a month | LargeConsec | weather | monthly | 32 km | North American Regional Reanalysis (NARR) |
| 1-month SPEI | SPEI | weather | 1-month | 0.5° | Global SPEI database |

**Fuel variables**

| | | | | | |
|---|---|---|---|---|---|
| Monthly mean Leaf Area Index (LAI) | LAI | fuel | monthly | 500 m | MODerate resolution Imaging Spectroradiometer (MODIS) |
| Monthly mean sum of neighboring LAI | convLAI | fuel | monthly | 500 m | MODerate resolution Imaging Spectroradiometer (MODIS) |
| Monthly mean soil moisture at 0-10 cm | soil | fuel | monthly | 0.125° | North American Land Data Assimilation System (NLDAS-2) |

**Geospatial and population variables**

| | | | | | |
|---|---|---|---|---|---|
| Land types | land_ | fix | | 30 m | National Land Cover Database (NLCD) |
| Ecoregion types | eco_ | fix | | | U.S. Environmental Protection Agency (EPA) |
| Population density | pop | fix | | | U.S. Census 2010 |

**Climate variables (over 1979-2000)**

| | | | | | |
|---|---|---|---|---|---|
| Long-term average and standard deviation of monthly temperature | temp_avg; temp_sd | fix | monthly | 32 km | North American Regional Reanalysis (NARR) |
| Long-term average and standard deviation of monthly mean of daily precipitation | apcp_avg; apcp_sd | fix | monthly | 32 km | North American Regional Reanalysis (NARR) |
| Long-term average and standard deviation of monthly maximum temperature | tmax_avg; tmax_sd | fix | monthly | 32 km | North American Regional Reanalysis (NARR) |
| Long-term average and standard deviation of monthly total precipitation | asum_avg; asum_sd | fix | monthly | 32 km | North American Regional Reanalysis (NARR) |

| Climate anomalies of monthly mean temperature | temp_anomaly | climate | monthly | 32 km | North American Regional Reanalysis (NARR) |
|---|---|---|---|---|---|
| Climate anomalies of monthly mean of daily precipitation | apcp_anomaly | climate | monthly | 32 km | North American Regional Reanalysis (NARR) |
| Climate anomalies of monthly mean RH | rhum_anomaly | climate | monthly | 32 km | North American Regional Reanalysis (NARR) |
| Climate anomalies of monthly maximum temperature | tmax_anomaly | climate | monthly | 32 km | North American Regional Reanalysis (NARR) |
| Climate anomalies of monthly minimum RH | rmin_anomaly | climate | monthly | 32 km | North American Regional Reanalysis (NARR) |
| Climate anomalies of monthly total precipitation | asum_anomaly | climate | monthly | 32 km | North American Regional Reanalysis (NARR) |

**Lagged variables**

**Winter-spring fire season**

| The monthly mean of daily precipitation of months *t-1* | apcp_mean1m | weather | monthly | 32 km | North American Regional Reanalysis (NARR) |
|---|---|---|---|---|---|
| The average SPEI of the months *t-1*, *t-1* to *t-2*, *t-1* to *t-3*, *t-1* to *t-4*, *t-1* to *t-5*, and *t-1* to *t-6* | SPEI_mean1m | weather | monthly | 0.5° | Global SPEI database |
| The averages of LAI and sum of neighboring LAI for the months *t-1* to *t-6* | LAI_mean6m, convLAI_mean6m | fuel | monthly | 500 m | MODerate resolution Imaging Spectroradiometer (MODIS) |

**Summer fire season**

| The average of monthly mean of daily precipitation for months *t-1, t-1* to *t-2* | apcp_mean1m | weather | monthly | 32 km | North American Regional Reanalysis (NARR) |
|---|---|---|---|---|---|
| The average of monthly mean temperature for months *t-1* and *t-1* to *t-2* | temp_mean1m | weather | monthly | 32 km | North American Regional Reanalysis (NARR) |
| The average of SPEI of months *t-1*, *t-1* to *t-2*, and *t-1* to *t-3* | SPEI_mean1m | weather | 1-month | 0.5° | Global SPEI database |


**Table 2.** Model performance at grid level for the two fire seasons.

| Fire season | Evaluation Metrics |
|---|---|

| | Accuracy | Recall | Precision | F1-score | AUC | $R^2$ | RMSE (km$^2$) | MAE (km$^2$) |
|---|---|---|---|---|---|---|---|---|
| F1 | 0.74 | 0.88 | 0.73 | 0.79 | 0.82 | 0.42 | 8.37 | 1.13 |
| F2 | 0.74 | 0.84 | 0.71 | 0.77 | 0.83 | 0.40 | 4.26 | 0.57 |



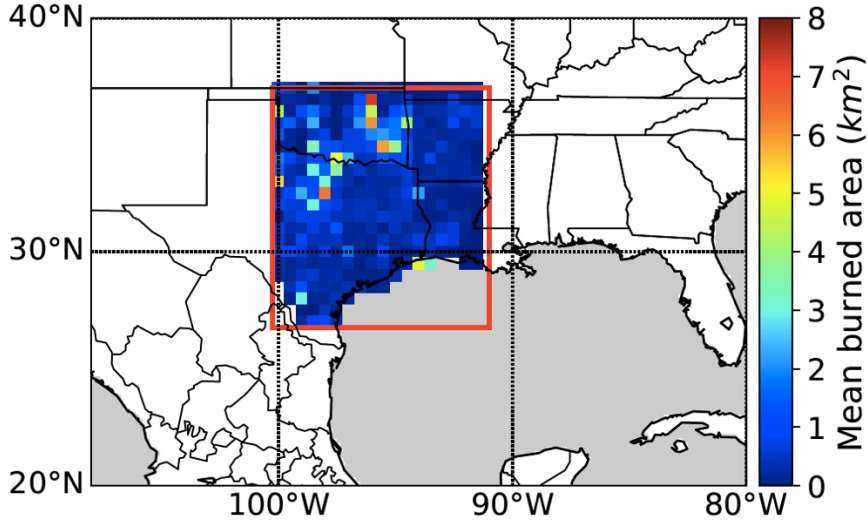

**Figure 1.** The colored grid boxes show the averaged burned area for the winter-spring and summer fire seasons during 2002-2015 from Fire Program Analysis Fire-Occurrence Database (FPA-FOD). The red box denotes the South Central US domain.


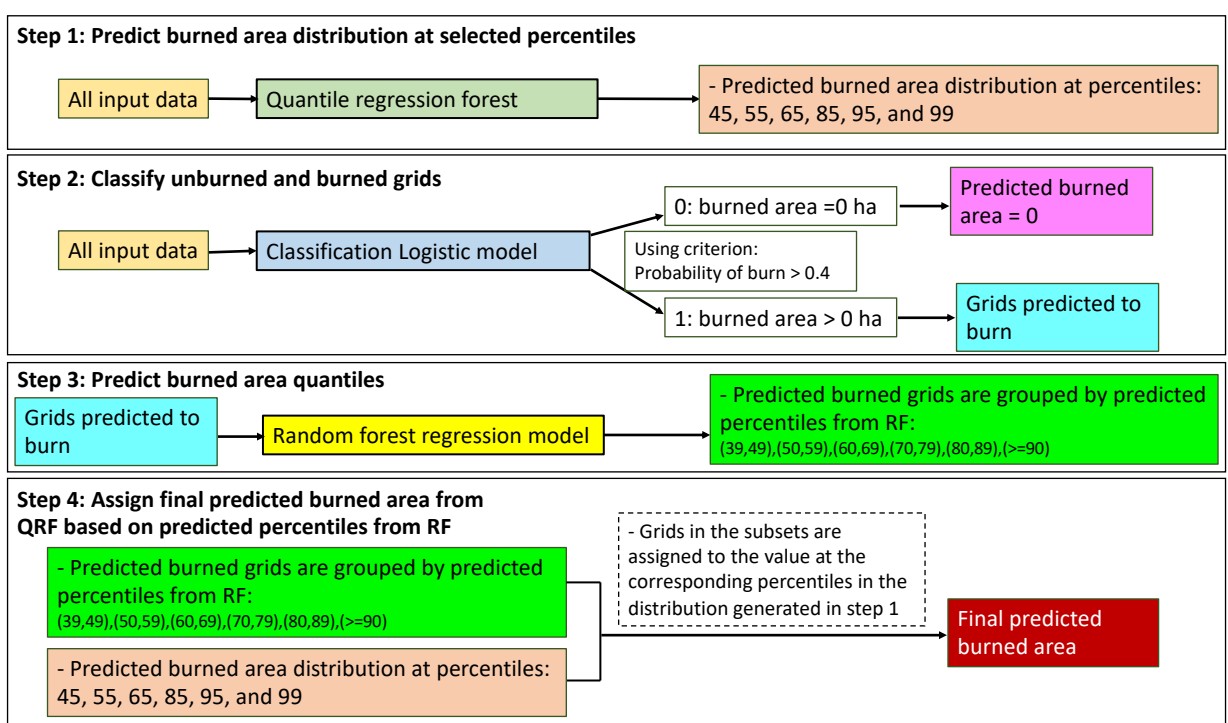


**Figure 2.** Illustration of the steps in the developed model. The model includes four steps and three machine learning algorithms, including a logistic model (dark blue) classifying a grid with non-zero burned area or not, a random forest model (yellow) predicting percentiles of burned area, and a quantile regression forest (dark green) predicting conditional burned area distributions.


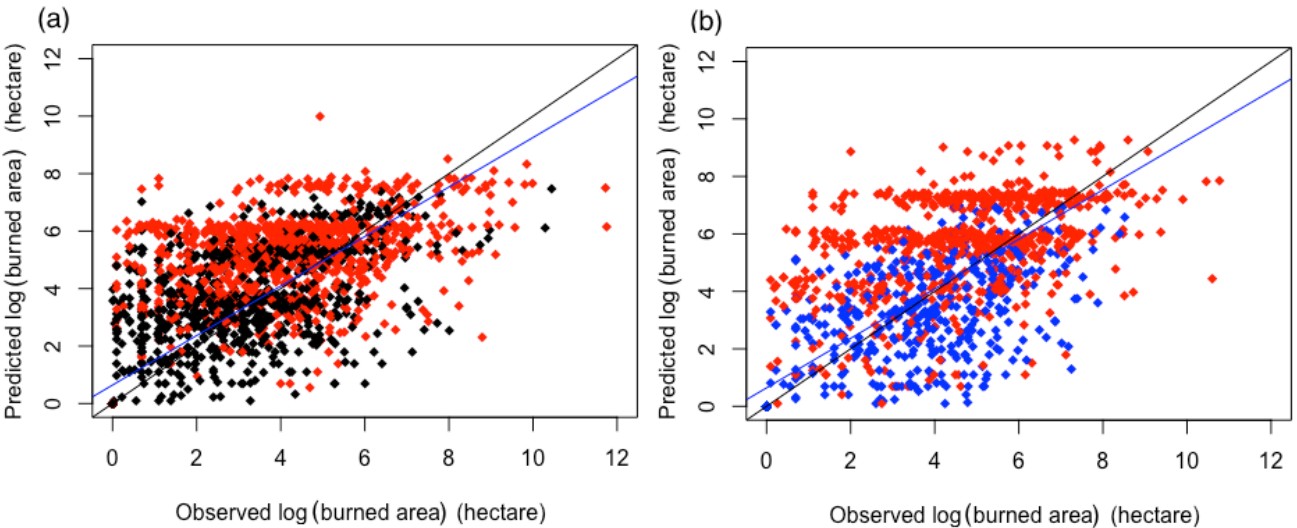

**Figure 3.** Comparison between observed and predicted logarithmic burned area (hectare) for the (a) winter-spring and (b) summer fire season in selected years: 2011 (red, year of the largest burned area), 2008 (blue, year with burned area close to the 14-year mean of its season), and 2014 (black, year with burned area close to the 14-year mean of its season). The black line represents the line of unity and the blue line is a best fit to the data by linear regression.

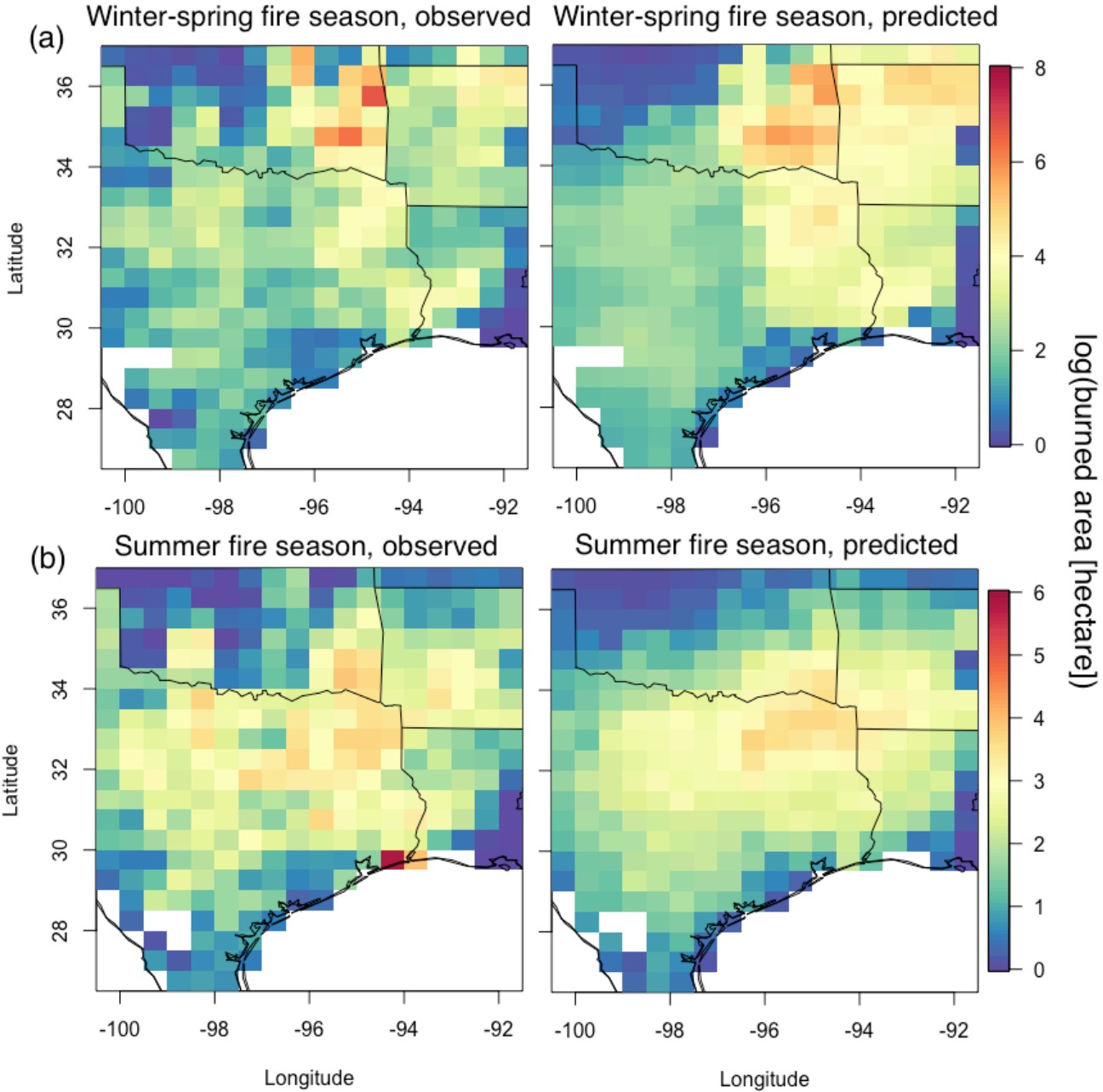


**Figure 4.** Map of monthly mean observed and predicted burned area averaged from 2002 to 2015 for the (a) winter-spring and (b) summer fire season.

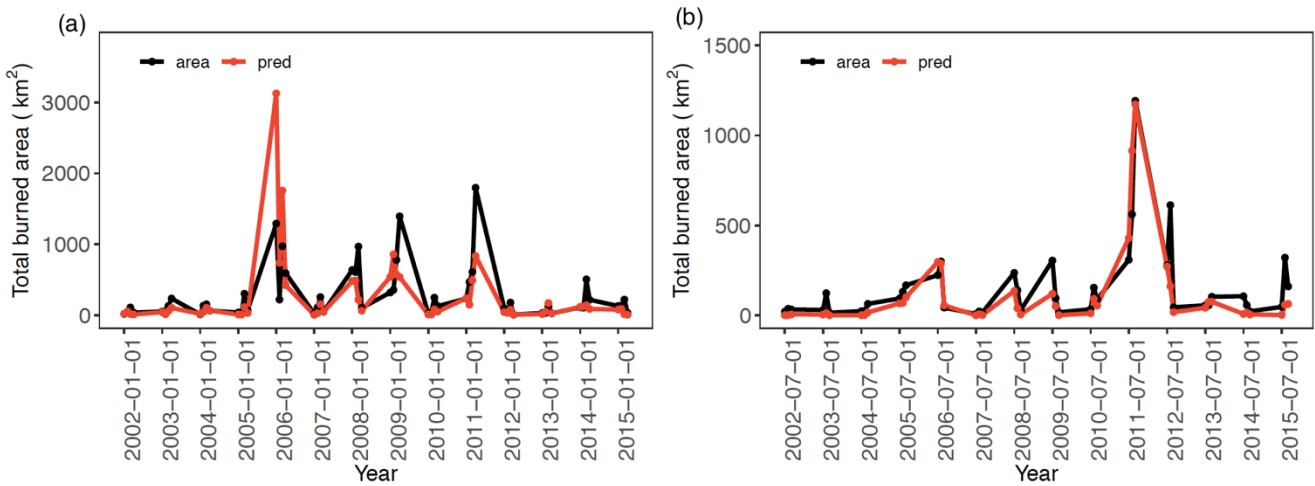

**Figure 5.** Timeseries of observed (black line) and predicted total burned area (red line) over South Central US for the (a) winter-spring and (b) summer fire season.

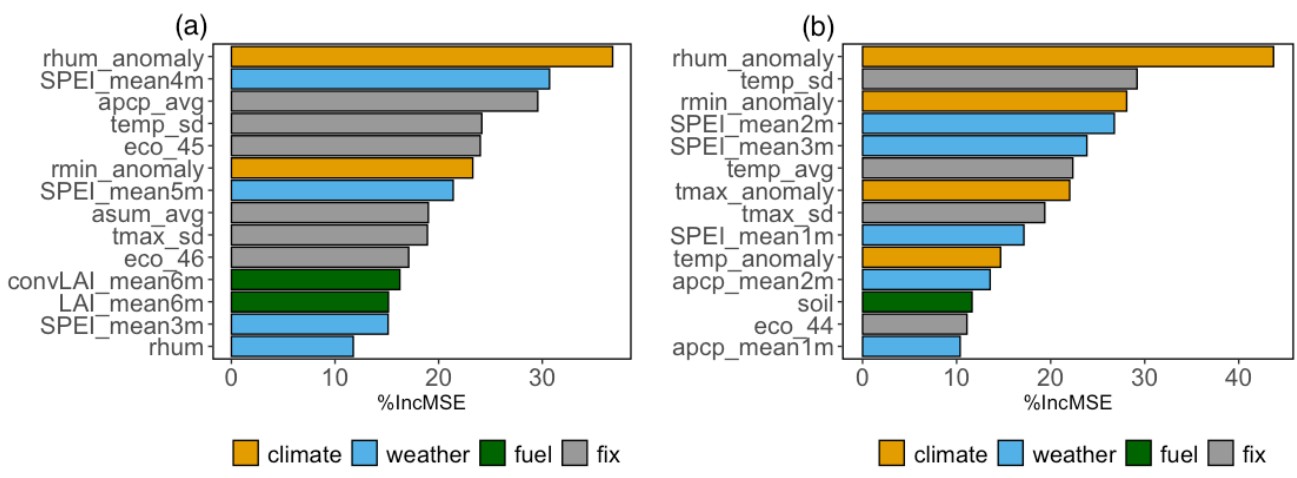

**Figure 6.** Relative importance of the top 14 variables presented by increase in mean square errors (%IncMSE) for (a) the winter-spring fire season (b) summer fire season.

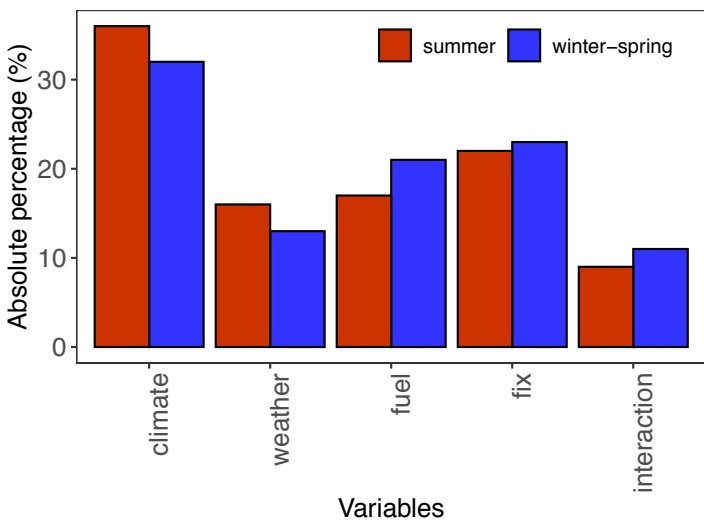

**Figure 7.** The mean scaled absolute percentage of the environmental controls for the winter-spring (blue) and summer fire
season (red).