# Peer review of "Quantifying the effects of environmental factors on wildfire burned area in South Central US using integrated machine learning techniques"

_Atmospheric Chemistry and Physics, 2019_

## Referee Comment (RC1) · Anonymous Referee #2 · 27 Nov 2019

Summary: This manuscript described a method for estimating burned area in the southern central region of the United States using three machine learning methods applied serially, with training derived from an existing dataset. The results show some skill in modeling total burned area over large areas. The work is focused mainly on the role of climatic variables in estimating burned area totals. While the methods in this paper might be of interest to the broader community, the manuscript is not well written (the structure is difficult to follow and it requires significant language editing throughout), the results cannot be reproduced because there is not enough information about the input variable processing, and the significance and limitations of the study are not explained well. Again, this method could prove to be useful to the broader community,

but the manuscript needs significant work and for that reason I recommend rejecting this paper.

General Comments: While I think the methods presented in the manuscript have potential to produce useful results, the manuscript needs to be improved in order to create a more logical flow of information, better describe the data used, illustrate the output, provide a more complete literature review, and provide details about the usefulness and limitations of this study. Furthermore, it requires editing beyond the scope of scientific peer-review.

The goals of this study are unclear – is the goal to predict wildfires in the future based on weather conditions, to support climate projections, or to simply estimate the amount of burned area? A related critique is that the structure of the paper makes it difficult for the reader to follow, there are effectively two methods sections with the results of the first set of methods in the middle. Additionally, the authors never present a figure showing the modeled burned area, which should be the main output of this work and really needs to be emphasized in the main body of the manuscript.

The authors have not considered a large body of wildfire research regarding satellite observations-driven modeling which is relevant to this work in the background research. Similar studies involving the effects of climate on total burned area should be noted by the authors, including Andela et al., 2017 and Zubkova et al., 2019. Additionally, the methods section refers to aspects of the data which are not described until a later section. This organization is difficult for the reader to follow, and the description of the data used is insufficient, in part because the sources of the input data are not provided. The data preprocessing methods are unclear as well – how is a discrete thematic variable like land cover type represented at 0.5-degree resolution? How are the translations between quantiles and area being made, given that the area of the grid cell varies with latitude? Also, the authors say the model predicts burned area at 50 km spatial resolution (with no indication of the map projection used), this is not the same as 0.5 degrees and this discrepancy needs to be resolved. There are also questions

about the fire data used to train the model – are prescribed fires included in the data (by definition, these are not wildfires in most cases)? Is there an estimate of the number of fires which are omitted? Given the quantile-based approach, what happens if there is a fire or amount of burning which is greater than any in the training dataset (i.e. it would fall out of the range of the training data unless there was a training cell with 100% burned area)? Is the length of the training dataset long enough to capture all variability in fire activity as it relates to climatic conditions? Why were remote sensing-derived datasets not considered?

An important aspect of fire regimes which was not adequately considered in the manuscript is the role of human activity in the fire regime, especially in the United States where humans play an active role in the fire regime through suppression, ignitions, fuel load management, and landscape fragmentation in addition to being the source of ignition of approximately 85% of fires (according to the US Forest Service). These effects vary as function of not only population density, but sociopolitical norms which can vary from state to state. Recent papers such as the Andela et al. 2017 paper claim human activity is the major control on fire activity, and as such it cannot be ignored in a study region where the fire regime is likely human-driven. While the datasets describing human activity are certainly far from perfect, it is not possible to describe fire activity in a human-driven fire regime without considering human influences.

Finally, there needs to be more effort in describing the expected impact of the work and the limitations of the method. For example, the abstract mentions that the work can be used to assess fire management strategies but provides no details on how or why. The quality of the input data is not discussed, which will propagate errors through the model, as well as the serial structure of the integrated model itself. At present, the manuscript is too focused on the machine learning exercise rather than on the scientific value of the work.

Specific Comments:

L75: Why were other months of the year excluded?

L81: "Uneven data" is used throughout the paper but is not defined. Does this refer to unevenness spatially, temporally, or both?

L92-95: Given that the model compares the output to the quantile ranges, is it capable of estimating an amount of burning greater than has been observed in the training data?

L155: Is there any concern about propagation of error through the model? What is the benefit of running three models in serial rather than one model alone or several models in an ensemble?

L164: Is there an estimate of the number of fires missed? Small fires constitute most of the fires by number, even though they add up to relatively little burned area (e.g. Malamud, Millington, and Perry 2005). It is noted that the dataset omits most small fires occurring on private land – these are not generally wildfires and such fires should be omitted anyways if the study is about wildfires.

L194-195: I don't think the climatic variables can be considered as fixed, especially since the assumption in later parts of the paper surround climate change scenarios which means their values do vary through time.

L209: Is any consideration given to preventing overfitting due the correlation between variables? For example, ecoregions and landcover types are likely to be related to one another.

L232: Please clarify the phrase "horizontal scale of around 700 x 700 km2" – the use of horizontal scale implies a one-dimensional unit (length) which does not match the unit specified. Also, as a suggestion, 700 x 700 km2 seems ambiguous and could be more clearly represented as "700 km x 700 km" or "490,000 km2"

L252: SUS is never defined

L268-270: One could argue that the model is in fact "hardwired" (editorially, the term is jargon and should be replaced) to the geographical features of the study domain – geography deals with the human components of space and time as well as the physical components. The tendency of the human population to ignite or suppress fires as a result relationship to sociopolitical factors (like local regulations) will influence the fire regime in ways which will not be captured by climatic variables and will change from location to location.

L286: Why were 14 variables chosen? This seems like an arbitrary cutoff, especially given the large number of variables which went into the model.

L306: The fuel-related variables are among the least important presented in Figure 5 – how can the conclusion be drawn that fuel abundance is what determines the amount of burned area?

Table 1: The resolution of the data is presented, but it's not clear how the data are being re-gridded to the working resolution - if the fire analysis is being done at 0.5 degree and the climate data is at 32 km resolution then there are < 4 cells per burned area data point and the way which those ∼4 cells are represented has significant consequences.

---

## Referee Comment (RC2) · Anonymous Referee #1 · 3 Dec 2019

**Title: Predicting wildfire burned area in South Central US using integrated machine learning techniques**

This paper proposed a machine learning method to predict gridded monthly wildfire burned area during 2002-2015 over the South Central United States and identify the relative importance of the predicted factors on the burned area for both the winter-spring and summer fire seasons. The method is able to alleviate the problem of unevenly-distributed burned area data due to the grid-level resolution. The result is interesting and constructive to some extent.

The authors said the machine learning method can achieve the R2 value of ~0.4. However, this result is hard to say it as a high accuracy. The authors can consider to compare more machine learning algorithms, such as AdaBoost, XGBoost. The low accuracy will also affect the reliability of the importance of predicted factors.

Therefore, I would recommend a major revision.

➢ P1L25: have been seen increasing?

➢ P2L51: You said the machine learning methods were used to estimate total burned area aggregated over a large-scale domain in past studies. In this study, you focus on the grid-level resolution. Could you describe what the resolution of past studies is? Do these works have the issue of the unbalanced distributed burned area?

➢ P3L83: The small fire is less than 10 ha, and the large fire is greater than 100 ha. However, the small fire is defined as less than 25 ha, and the large fire is greater than 150 ha in P2L58. Could you explain why they are different?

➢ I don't think Figure 2 looks nice and it should be re-organized better. For example, the arrow between step 1 and step 2 confuses that there may be an input-output relationship. In fact, is it correct that they are independent processes?

➢ P3L90: The description of the four steps is not very clear. This paragraph should be rewritten. Is it correct the quantiles are the x-axis of frequency histogram? Why do you choose these quantiles? Will the pre-defined parameters induce uncertainties?

➢ P3L90: Although the authors claim that the four steps method will alleviate the problem of uneven-distributed dataset, the multi-steps will introduce some risks. For example, if the second step wrongly classifies the burned area as the non-burned area, the bias will be amplified because it won't enter into step 3.

➢ P4L126: Please explain the assumption or give the reference that grids with larger burned area will have more right-shifted burned area distribution than the

distributions of the training set.

➢ P6L171: Please add references.

➢ P8L223: Please add the F1-score performance criteria because you mentioned it in L235.

➢ P8L235: Could you please plot the AUC curve so that it could help to analyze the TP rate and FP rate. You can also analyze the F1-score performance by Precision and Recall. This will help to understand whether the classifier is underprediction or overprediction.

➢ P8L235: The performance accuracy of the classifier and the regressor in Table 2 is not very high. Typically, the F1-score of a good classifier can achieve over 0.8 and the RMSE of a good regressor is lower than 0.2. Could you compare your results with some other machine learning methods, such as Adaboost, XGBoost.

➢ What does "630" mean in Table 2?

➢ P8L247: Please add some references for past studies.

➢ P8L251: You compare your results with Chen et al. (2016) and Liu and Wimberly (2015). I wonder whether they are comparable if they are under different factors, different periods and different regions.

➢ Please explain the meaning of the blue line in Figure 3.

➢ P9L281: Although the importance of Random Forest help to identify some key factors, they depend on the accuracy of the machine learning method. If the accuracy is not very high, it will reduce the reliability of the information. On the other hand, the importance can't provide how the change trend of factors affect the prediction.

➢ P9L299: I can't find the reference "Westerling and Bryant (2008)" in the reference list.

➢ There are several obvious typos in the manuscript, and the English language is poor. I think the authors should be asked to have the manuscript proofread by a native English speaker before the article can be considered for publication in a scientific journal.

---

## Author Comment (AC1) · 11 Feb 2020

**Response to Reviews**

We thank the two reviewers for their constructive comments to improve the manuscript. Their comments are reproduced below with our responses in blue.

**Reviewer #1**

This paper proposed a machine learning method to predict gridded monthly wildfire burned area during 2002-2015 over the South Central United States and identify the relative importance of the predicted factors on the burned area for both the winter-spring and summer fire seasons. The method is able to alleviate the problem of unevenly-distributed burned area data due to the grid-level resolution. The result is interesting and constructive to some extent. The authors said the machine learning method can achieve the R2 value of ~0.4. However, this result is hard to say it as a high accuracy. The authors can consider to compare more machine learning algorithms, such as AdaBoost, XGBoost. The low accuracy will also affect the reliability of the importance of predicted factors. Therefore, I would recommend a major revision.

We appreciate the feedback from the reviewer. It is easily understood that the prediction accuracy of wildfire burned areas will increase as the spatial and temporal resolution of the prediction model decreases. In comparison to previous studies that predict wildfire burned areas, our work is unique in two aspects: (1) our prediction performance ($R^2$ of ~0.4) was reported at the 0.5° x 0.5°-grid scale rather than over an aggregated spatial domain; (2) we did not exclude unburned grids or small burned grids from the prediction. Since very few published studies had both of these two features, we can only compare our results with the prior studies predicting burned area at similar spatial scales as our work (see the Table R1 below). These studies have $R^2$ values all below 0.3, despite using a coarser spatial resolution (~100 km x 100 km) and sometimes also a coarser temporal resolution (annually). Considering the complexity of wildfires and intrinsic nature of unevenly distributed burned area, we argue that $R^2$ value of around 0.4 from our work is a significant improvement over those previously published studies. Furthermore, the machine learning approaches developed by our work were motivated by the need to reduce the uneven distribution of burned area data so as to achieve a higher prediction accuracy.

We tested other boosting methods as suggested by the reviewer and they did not achieve significantly better results (see our response to comment #11). In addition, at this stage we do not foresee physical explanations to adopt these boosting methods. To address the reviewer's concern, we have rewritten the comparison of our work with others in the revised manuscript and elaborated in more detail how much better our model is compared with previously published works.

Table R1. Studies using statistical methods to estimate burned area at spatial scales similar as this study

| Region | period | Method | Spatial scale | Temporal scale | $R^2$ | References |
|---|---|---|---|---|---|---|
| Spain | 1990-2008 | MARS | 25 km x 25 km ~100 km x 100 km | Monthly | Median correlation R=0.29 ($R^2$~0.08) | Bedia et al. (2014) |
| EU-Mediterranean | 1985-2011 | MLR | ~108 km x 108 km | Annual | Median $R^2$=0.28 | Urbieta et al. (2015) |
| Pacific western coast of USA | 1985-2011 | MLR | ~108 km x 108 km | Annual | Median $R^2$=0.22 | Urbieta et al. (2015) |

More detailed discussion is presented below in the specific comment #11.

Regarding the reliability of the inferred importance of predicted factors, our results are robust based on the optimized model with the best results and the given set of predictor variables. To verify this, we also conducted 50 times 10-fold cross-validation by randomizing the order of all the data each time. The ranks of the variables by the median values of %IncMSE are identical to the ranks of the variables in our initial results. This indicates that feature importance identified by the random forest model is reliable and stable. We have included the above discussions in the revised manuscript.

1. P1L25: have been seen increasing?
We have changed the sentence to "many regions of the world have experienced an increase in frequency and intensity of wildfires …" (line 27).

2. P2L51: You said the machine learning methods were used to estimate total burned area aggregated over a large-scale domain in past studies. In this study, you focus on the grid-level resolution. Could you describe what the resolution of past studies is? Do these works have the issue of the unbalanced distributed burned area?
The resolution of past studies ranges from 25 km x 25 km to 2000 km x 2000 km, as listed in Table S1. The spatial resolution of the phytoclimatic zones in Bedia et al. (2014) ranges from 25 km x 25 km to 100 km x 100 km; the European countries in Amatulli et al. (2013) had the scales ranging from 300 km x 300 km to 1000 km x 1000 km. We have added the spatial and temporal resolution of these two studies mentioned in the main text (line 50-54).

For these studies, depending on the spatial scale, temporal scale, and wildfire characteristics (fire frequency, intensity etc.), the burned area distribution can be normal-distributed or right-skewed. Generally, studies predicting annual burned area

on a country scale (spatial scale larger than 100 km x 100 km) do not have the issue of uneven-distributed burned area because the burned area are already aggregated. For example, Amatulli et al. (2013) shows the distributions of annual burned area for the countries in European Mediterranean (spatial scales ranging from 300 km x 300 km to 1000 km x 1000 km). For most of the countries, the burned area is normally distributed (Fig R1 below).

[Figure]

Fig. R1. Box plots of annual observed (left) and projected burned area in all study regions using the MARS models under B2 (middle) and A2 scenarios (right) for the European countries (adopted from Amatulli et al (2013); Fig. 8).

Another example of Carvalho et al. (2008) demonstrates that the distribution of burned area varies by districts in Portugal (spatial scales ranging from 25 km x 25 km to 100 km x 100 km). As Fig. R2 shows, the annual burned area distribution can be like a normal distribution for districts such as Braganca, or it can be very right-skewed for districts such as Evora and Portalegre. We also included the above example and associated discussions in the revised manuscript in line 322-325.

[Figure]

Fig. R2. Box plots of annual burned area and number of fires by Portuguese district for 1980-2004 period (adopted from Carvalho et al. (2008); Fig. 3b).

3. P3L83: The small fire is less than 10 ha, and the large fire is greater than 100 ha. However, the small fire is defined as less than 25 ha, and the large fire is greater than 150 ha in P2L58. Could you explain why they are different?

The definition of small fires at 25 ha and large fires at 150 ha was based on Steel et al. (2015) while the criterion of 10 ha was according to Yue et al. (2013). To avoid confusion and ensure the consistency, we changed the definition in line 183-186 from "10 ha" to "25 ha" and "100 ha" to "150 ha" and updated the statistics.

4. I don't think Figure 2 looks nice and it should be re-organized better. For example, the arrow between step 1 and step 2 confuses that there may be an input-output relationship. In fact, is it correct that they are independent processes?

We agree with the reviewer's concern and suggestions. We have re-organized and updated Figure 2, as shown below. Step 1 includes a quantile regression forest and step 2 includes a logistic regression with the same set of input variables. These two steps are independent processes but they are followed the order of the listed steps.

[Figure]

Fig. R3. Illustration of the steps in the developed model. The model includes four steps and three machine learning algorithms, including a logistic model (dark blue) classifying a grid with non-zero burned area or not, a random forest model (yellow) predicting quantiles of burned area, and a quantile regression forest (dark green) predicting conditional burned area distributions. (This figure is now Fig. 2. In the revised manuscript)

5. P3L90: The description of the four steps is not very clear. This paragraph should be rewritten. Is it correct the quantiles are the x-axis of frequency histogram? Why do you choose these quantiles? Will the pre-defined parameters induce uncertainties? (1) Yes, the quantiles represent to the position of the predicted burned area in the cumulative distribution of all the burned area data. For example, Fig. R4 shows the empirical cumulative distribution functions of the burned area in the winter-spring fire season. The value of quantile at 0.7 can be determined by the value on the x axis, which is 3.74 (log of hectares). To better clarify the idea of quantiles, we have replaced 'quantile' with 'percentile' in the manuscript and rewrote the definition of percentile in line 192-194.

[Figure]

Fig. R4. The empirical cumulative distribution functions of the burned area in the winter-spring fire season. The x axis is the log of burned area and the y axis is the cumulative probability. The red lines here point the value of burned area at quantile 0.7.

(2) The quantiles were chosen to include the whole conditional distribution. The first three quantiles were selected to represent the median values between the lower and upper bounds for the first three subgroups in step 3. The last three quantiles (0.85, 0.95, and 0.99) were chosen based on the assumption that grids with larger predicted burned areas (predicted quantiles > 0.70) in the testing set will have burned area distributions that are more right-shifted than the distribution of the whole training set (Fig. R5). The quantiles were selected to reduce the model bias at the high end of burned areas. We have edited and added the above-mentioned explanations and Fig. R5 into the manuscript in line 199-205. In the revised manuscript, we have replaced 'quantile' with 'percentile'.

[Figure]

Fig. R5. Probability distribution of burned area for 10 folds of the training set (black line), testing set predicted to have percentiles less than 70 (blue), between 70 and 79 (yellow), between 80 and 89 (green), and equal to or larger than 90 (purple). (This figure is now Fig. S3. in the revised manuscript)

(3) To test the uncertainties that may be introduced by the pre-defined parameters (i.e. quantiles and subgroups), we switched the pre-defined quantiles but fixed the subgroups in the first sensitivity experiment. In this experiment, the last three quantiles were changed to the median values between a new set of lower and upper bounds, (0.75, 0.85, 0.95). As the Table R2 shows, the change of the quantiles has little effect on the overall MAE but affects the prediction of large burned areas with a smaller standard deviation in predicted values and larger MAE. Then we designed the second experiment by changing the number of subgroups, their ranges, and the corresponding quantiles. Changing both subgroups and quantiles has a marginal impact on MAE, although the standard deviation of the prediction is smaller than the results with the chosen quantiles and subgroups.

Generally, changing pre-defined parameters has little effect on overall MAE for the two fire seasons but the MAE of large burned area is larger and standard deviation of the predicted values becomes smaller. The pre-defined parameters may lead to uncertainties mostly affect the spread of the predictions and the magnitudes of large burned areas. Despite the sensitivities, the prediction model with the chosen quantiles is able to predict burned area at 0.5° x 0.5°-grid scale and achieve higher prediction accuracy compared to prior studies. Table R2 and the above discussions have been included into the manuscript (line 528-537).

Table R2. Comparison of MAE, MAE of large burned area, and standard deviation of predictions between the model with the chosen quantiles, quantile test 1, and quantile test set2 (This table is now Table S7 in the revised manuscript)

| Model | With the chosen quantiles* | Quantile test set 1* | Quantile test set 2* |
|---|---|---|---|
| MAE (log of area; winter-spring) | 1.37 | 1.30 | 1.29 |
| MAE of large burned area[†] (log of area; winter-spring) | 2.13 | 2.64 | 2.81 |
| Standard deviation of predictions (log of area; winter-spring) | 2.42 | 2.09 | 2.09 |
| | | | |
| MAE (log of area; summer) | 1.17 | 1.12 | 1.11 |
| MAE of large burned area[†] (log of area; summer) | 2.25 | 2.42 | 2.52 |
| Standard deviation of predictions (log of area; summer) | 2.19 | 1.93 | 1.92 |

* Model developed in this study: Use the selected quantiles of 0.45, 0.55, 0.65, 0.85, 0.95, and 0.99 and six subgroups of (0.39, 0.49), (0.50, 0.59), (0.60, 0.69), (0.70, 0.79), (0.80, 0.89), (>=0.90).
* Set 1: Use the selected quantiles of 0.45, 0.55, 0.65, 0.75, 0.85, and 0.95 and six subgroups of (0.39, 0.49), (0.50, 0.59), (0.60, 0.69), (0.70, 0.79), (0.80, 0.89), (>=0.90).

* Set 2: Use the selected quantiles of 0.475, 0.63, 0.78, and 0.93 and four subgroups of (0.39, 0.55), (0.56, 0.70), (0.71, 0.85), (0.86, 1.00).

† Large burned area here is defined as the burned area larger than $90^{th}$ percentile.

6. P3L90: Although the authors claim that the four steps method will alleviate the problem of uneven-distributed dataset, the multi-steps will introduce some risks. For example, if the second step wrongly classifies the burned area as the nonburned area, the bias will be amplified because it won't enter into step 3.

The reviewer is correct that biases from one step could be propagated to the subsequent steps, for example when the burned grids are predicted not to burn or when the unburned grids are predicted to burn. For the first case, when the burned grids are incorrectly predicted not to burn, the low bias is introduced because the burned grids would not proceed to step 3. For the second case, inclusion of unburned grids in step 3 may introduce a positive bias. We have included the discussions of the error propagation in section 6 (line 511-513).

To demonstrate our four-step model can achieve a higher accuracy and alleviate the issue of uneven-distributed dataset, we compare the prediction performance using random forest alone with that of the four-step model developed in this study, as shown in the Table R3 below:

Table R3. Comparison of MAE and skewness between the RF model and the developed model (The information of this table is now included into the Table S2 in the revised manuscript)

| Model | RF alone | Model developed in this study |
|---|---|---|
| MAE (winter-spring) | 1.34 | 1.13 |
| Skewness (winter-spring) | 37.40 (burned area) | 0.70 (quantiles) |
| | | |
| MAE (summer) | 0.70 | 0.57 |
| Skewness (summer) | 33.83 (burned area) | 0.96 (quantiles) |

Skewness is a measure of the asymmetry of the probability distribution of a random variable about its mean. The skewness of a random variable X is the third standardized moment $\widetilde{\mu_3}$, defined as:

$$\widetilde{\mu_3} = E\left[\left(\frac{X-\mu}{\sigma}\right)^3\right] = \frac{\mu_3}{\sigma^3} = \frac{E[(X-\mu)^3]}{(E[(X-\mu)^2])^{3/2}} = \frac{\kappa_3}{\kappa_2^{3/2}}$$

where $\mu$ is the mean, $\sigma$ is the standard deviation, E is the expectation operator, $\mu_3$ is the third central moment, and $\kappa_t$ are the $t$-th cumulants. If skewness is less than -1 or greater than +1, the distribution is highly skewed. If skewness is between -1 and -0.5 or between +0.5 and +1, the distribution is moderately skewed. If skewness is between -0.5 and 0.5, the distribution is approximately symmetric. The positive value indicates that the tail is on the right side of the distribution while negative value indicates that the tail is on the left.

Our model has a lower MAE, by 15% and 19% for the winter-spring and summer fire season, respectively, compared to the single RF model. The distribution of the quantiles in the developed model is more uniform than the distribution of the burned area, as shown in Figure R6 below and the skewness. We have added a discussion of this issue in the text (line 355-365) and included Table R3 in the supplementary information. The information of skewness calculation has been included in the supplementary information. Note that we have replaced 'quantile' with 'percentile' in the manuscript to better clarify the idea of quantiles.

[Figure]

Fig. R6. (a) Histogram of burned area (b) Histogram of the percentile groups of burned area for the winter-spring fire season. (This figure is now Fig. S2. In the revised manuscript)

7. P4L126: Please explain the assumption or give the reference that grids with larger burned area will have more right-shifted burned area distribution than the distributions of the training set.
The above-mentioned Figure R5 shows the burned area distributions of training sets and testing sets categorized by predicted quantile groups. Grids that are predicted to have larger burned area (predicted quantiles larger than 0.70) have more right-shifted burned area distributions compared to the distribution of the training set. We have included the figure and explained the assumption in the manuscript (line 199-205 and Fig S3).

8. P6L171: Please add references.
We have added references in line 106-108.

9. P8L223: Please add the F1-score performance criteria because you mentioned it in L235.
We have added the description of F1-score in line 267-270.

10. P8L235: Could you please plot the AUC curve so that it could help to analyze the TP rate and FP rate. You can also analyze the F1-score performance by Precision

and Recall. This will help to understand whether the classifier is underprediction or overprediction.

The ROC curves are included in the supplementary (Fig S4), as demonstrated below.

[Figure]

Fig. R7. The ROC curve of burning classification for the (a) winter-spring and (b) summer fire season. (This figure is now Fig. S4. In the revised manuscript)

The ROC curves show good performance of the models, given that the ROC curves are toward the upper left corner and the AUC for the two fire seasons are 0.82 and 0.83. The accuracy and F1-score are 0.74 and 0.79 for the winter-spring fire season. For the summer fire season, the accuracy and F1-score are 0.74 and 0.77. The above results demonstrate the model ability of predicting burned grids with the optimal balance of recall and precision. The values of AUC, recall, precision, and F1-score are also updated in Table 2. We have included the results and discussions of model performance in the main text (line 292-298).

11. P8L235: The performance accuracy of the classifier and the regressor in Table 2 is not very high. Typically, the F1-score of a good classifier can achieve over 0.8 and the RMSE of a good regressor is lower than 0.2. Could you compare your results with some other machine learning methods, such as Adaboost, XGBoost.

We compared our classification model results with some other machine learning methods (the parameters of each model have been optimized):

Table R4. Comparison of accuracy, recall, precision, and F1-score between the logistic regression model, RF model and XGBoost model

| Model | Logistic regression (model for this study) | Random forest | XGBoost |
|---|---|---|---|
| Winter-spring fire season | | | |
| **Accuracy** | 0.74 | 0.82 | 0.80 |
| **Recall** | 0.88 | 0.86 | 0.86 |
| **Precision** | 0.73 | 0.83 | 0.80 |
| **F1-score** | 0.79 | 0.84 | 0.83 |
| Summer fire season | | | |
| **Accuracy** | 0.74 | 0.81 | 0.81 |
| **Recall** | 0.84 | 0.82 | 0.82 |
| **Precision** | 0.71 | 0.82 | 0.81 |
| **F1-score** | 0.77 | 0.82 | 0.81 |

RF and XGboost show better performance in terms of accuracy, precision, and F1-score. However, they identify less burned grids (value of Recall) and require more parameter tuning and runtime, compared to logistic model. Considering the comparable performance, more identified burned grids, less model tuning, and less runtime, we chose logistic model as our classification model because it can be simply applied to different regions which is a competitive advantage for future applications of the prediction model.

As for the burned area prediction, we've compared our results with RF model in question 6. Here we included the model results from XGBoost, as it shows below:

Table R5. Comparison of MAE and skewness between the developed model, RF model, XGBoost model (This table is now Table S2 in the revised manuscript)

| Metrics | Model developed in this study | RF alone | XGBoost alone |
|---|---|---|---|
| MAE (winter-spring) | 1.13 | 1.34 | 1.26 |
| Skewness (winter-spring) | 0.70 (quantiles) | 37.40 (burned area) | 37.40 (burned area) |
| | | | |
| MAE (summer) | 0.57 | 0.70 | 0.67 |
| Skewness (summer) | 0.96 (quantiles) | 33.83 (burned area) | 33.83 (burned area) |

Our four-step model has a lower MAE, which deceases by 11% and 15% for the winter-spring and summer fire season, respectively, compared to the XGboost model. The

developed model shows better performance in predicting burned area, compared to using RF or XGboost model. The model results from XGboost were also included in the Table S2 and corresponding discussion in the manuscript (line 355-365).

12. What does "630" mean in Table 2?
We have removed the misplaced line number 630 from Table 2.

13. P8L247: Please add some references for past studies.
The comparison of model performance to the previous studies has been rewritten and the associated references can be found in the manuscript (line 299-330).

14. P8L251: You compare your results with Chen et al. (2016) and Liu and Wimberly (2015). I wonder whether they are comparable if they are under different factors, different periods and different regions.
Since there are very few studies predicting gridded burned area directly and among them there is no study focusing on the South Central US, we chose to compare our results with these studies in terms of the approaches (i.e. excluding unburn grids or not), the temporal and spatial resolution, and the percent of variance explained by the model (i.e. R-square), regardless of their study regions, periods, and used predictors.

Although the study regions, study period, and the used predictors in this study are different from prior studies mentioned for comparison in the main text, the developed model in this study demonstrates some advantages compared to other models. First, both Chen et al. (2016) and Liu and Wimberly (2015) excluded unburned or small-burned grids when building their models, thus failing to capture the response of small fires size to predictor variables. Second, both studies focused on annual burned area in a spatial resolution of 1°x 1°, while the spatial and temporal resolution of our four-step model is finer both spatially and temporally (0.5° x 0.5° and monthly burned area). Our four-step model is able to resolve the fire-predictor relationship in a seasonal and a relatively-finer spatial scale.

We also included other studies with a similar spatial resolution for comparison. Urbieta et al. (2015) used multiple linear regression (MLR) to predict annual burned area of provinces and national forests in the southern countries of the European Union (EUMED) and Pacific Western US (PWUSA) (spatial resolution ~ 108 km x 108 km). For all the provinces/national forests, the median $R^2$ is 0.28 for EUMED and 0.22 for PWUSA. Carvalho et al. (2008) utilized MLR to predict monthly burned area of Portuguese districts (spatial resolution ranging from 25 km x 25 km to 100 km x 100 km) and their $R^2$ range from 0.43 to 0.80. Even though they achieved a better model performance for some districts, their models had poorer performance for the districts with very right-skewed burned area distribution (Figure R2 shown above), including Evora ($R^2$=0.43), Portalegre ($R^2$=0.45). Another example of Bedia et al. (2014) predicted monthly burned area of phytoclimatic zones in Spain (~ 25 km x 25 km to

100 km to 100 km) by using multivariate adaptive regression splines (MARS) and they obtained R² ranging from 0.01 to 0.37.

Although the model performance may vary depending on regions, fire characteristics, time scales, and predictors, the R² value of around 0.4 that we achieved to predict monthly burned area at a spatial resolution of 0.5°x0.5° is a significant improvement over previously published studies for burned area prediction at such spatiotemporal scale and the improvement was resulted from our efforts to alleviate the issue of unevenly-distributed burned area. We have rewritten the paragraphs to better explain the comparisons and the advantages of our models in line 299-330.

15. Please explain the meaning of the blue line in Figure 3.
The blue line is a best fit to the data by linear regression. We have added the descriptions of the blue line in the caption of Figure 3.

16. P9L281: Although the importance of Random Forest help to identify some key factors, they depend on the accuracy of the machine learning method. If the accuracy is not very high, it will reduce the reliability of the information. On the other hand, the importance can't provide how the change trend of factors affect the prediction.
(1) Based on the optimized model with the best results and the given set of predictor variables, the variable importance of predictors is reliable. Besides model accuracy, stability of the variable importance should also be considered (Han et al., 2012; He and Yu, 2010). To further ensure the variable ranking is stable, we conducted 50 times 10-fold cross-validation by randomizing the order of all the data each time. Figure R8 below shows the distributions of %IncMSE of each variable ranked by the median %IncMSE. Even though the feature importance varies a lot in different runs, the ranks by median values are identical to the variable ranks in our initial results, indicating the feature importance identified by the random forest model is stable. We have included the above discussions in the revised manuscript (line 382-385).

[Figure]

Fig. R8. Box plots of variable importance in %IncMSE from the 50 times 10-fold cross validation for (a) winter-spring and (b) summer fire season. (This figure is now Fig. S6. In the revised manuscript)

(2) Although the variable importance by RF cannot directly provide how the change trend of factors affect the prediction, like the coefficient in the linear regression, the partial dependence plots can be applied to the built model and show the marginal effect of a variable on the prediction performance (Friedman, 2001). The partial dependence plots consider a partial dependence function that is estimated by calculating averages in the training data and can be expressed as:

$$\widehat{f_{xS}}(xS) = \frac{1}{n} \sum_{i=1}^{n} \hat{f}(xS, x_C^{(i)}),$$

where xS is the feature we are interested in and $x_C^{(i)}$ are actual feature values for the features in which we are not interested. This partial function provides the average marginal effect on the prediction for given values of feature S.

Here we provide partial dependence plots for the burned area model and RH anomaly and mean SPEI of the preceding 4 months (the top two variables) for the winter-spring fire season (Fig. R9). For these two variables, there is a significant drop of fitted burned area when RH anomaly is larger than -1 and mean SPEI of the preceding 4 months larger than -0.6. The partial dependence plots demonstrate how change of a variable affects the predicted burned area. We have included the information and the above-mentioned examples in the revised manuscript (line 405-410).

[Figure]

Fig. R9. Partial dependence plots for the burned area model and (a) RH anomaly and (b) mean SPEI of the preceding 4 months for the winter-spring fire season. (This figure is now Fig. S7. In the revised manuscript)

17. P9L299: I can't find the reference "Westerling and Bryant (2008)" in the reference list.

We have included the reference "Westerling and Bryant (2008)" into the reference list.

18. There are several obvious typos in the manuscript, and the English language is poor. I think the authors should be asked to have the manuscript proofread by a native English speaker before the article can be considered for publication in a scientific journal.

We have revised the manuscript and the manuscript has been proofread by a native English speaker.

**References:**

Amatulli, G., Camia, A. and San-Miguel-Ayanz, J.: Estimating future burned areas under changing climate in the EU-Mediterranean countries, Science of The Total Environment, 450–451, 209–222, doi:10.1016/j.scitotenv.2013.02.014, 2013.

Bedia, J., Herrera, S. and Gutiérrez, J. M.: Assessing the predictability of fire occurrence and area burned across phytoclimatic regions in Spain, Natural Hazards and Earth System Sciences, 14(1), 53–66, doi:https://doi.org/10.5194/nhess-14-53-2014, 2014.

Carvalho, Flannigan, M., Logan, Miranda, A. and Borrego, C.: Fire activity in Portugal and its relationship to weather and the Canadian Fire Weather Index System, International Journal of Wildland Fire, 17, 328–338, doi:10.1071/WF07014, 2008.

Chen, Y., Morton, D. C., Andela, N., Giglio, L. and Randerson, J. T.: How much global burned area can be forecast on seasonal time scales using sea surface temperatures?, Environ. Res. Lett., 11(4), 045001, doi:10.1088/1748-9326/11/4/045001, 2016.

Friedman, J. H.: Greedy Function Approximation: A Gradient Boosting Machine, The Annals of Statistics, 29(5), 1189–1232, 2001.

Han, J., Kamber, M. and Pei, J.: Data Mining: concepts and techniques, Morgan Kaufmann., 2012.

He, Z. and Yu, W.: Stable feature selection for biomarker discovery, Comput Biol Chem, 34(4), 215–225, doi:10.1016/j.compbiolchem.2010.07.002, 2010.

Liu, Z. and Wimberly, M. C.: Climatic and Landscape Influences on Fire Regimes from 1984 to 2010 in the Western United States, PLOS ONE, 10(10), e0140839, doi:10.1371/journal.pone.0140839, 2015.

Steel, Z. L., Safford, H. D. and Viers, J. H.: The fire frequency-severity relationship and the legacy of fire suppression in California forests, Ecosphere, 6(1), 1–23, doi:10.1890/ES14-00224.1, 2015.

Urbieta, I. R., Zavala, G., Bedia, J., Gutierrez, J. M., San Miguel-Ayanz, J., Camia, A., Keeley, J. E. and Moreno, J. M.: Fire activity as a function of fire–weather seasonal severity and antecedent climate across spatial scales in southern Europe and Pacific western USA, Environmental Research Letters, 10(11), doi:10.1088/1748-9326/10/11/114013, 2015.

Yue, X., Mickley, L. J., Logan, J. A. and Kaplan, J. O.: Ensemble projections of wildfire activity and carbonaceous aerosol concentrations over the western United States in the mid-21st century, Atmos Environ (1994), 77, 767–780, doi:10.1016/j.atmosenv.2013.06.003, 2013.

**Reviewer #2**

Summary: This manuscript described a method for estimating burned area in the southern central region of the United States using three machine learning methods applied serially, with training derived from an existing dataset. The results show some skill in modeling total burned area over large areas. The work is focused mainly on the role of climatic variables in estimating burned area totals. While the methods in this paper might be of interest to the broader community, the manuscript is not well written (the structure is difficult to follow and it requires significant language editing throughout), the results cannot be reproduced because there is not enough information about the input variable processing, and the significance and limitations of the study are not explained well. Again, this method could prove to be useful to the broader community, but the manuscript needs significant work and for that reason I recommend rejecting this paper.

General Comments: While I think the methods presented in the manuscript have potential to produce useful results, the manuscript needs to be improved in order to create a more logical flow of information, better describe the data used, illustrate the output, provide a more complete literature review, and provide details about the usefulness and limitations of this study. Furthermore, it requires editing beyond the scope of scientific peer-review.

We thank the reviewer's comment. We have adjusted the sections of the manuscript to help readers better follow the article, and added more information about data, such as data source and the regridding method. With regards to the literature review, the focus of this study is on machine-learning-based prediction of wildfires and the relative importance of environmental controls of wildfires. Therefore, the literature review was mainly focused on this aspect. To follow the suggestions of the reviewer, we have added more literatures into the revised manuscript, including the ones mentioned by the reviewers. The details of the expected impact (line 70-73 and 558-567) and the limitation of this study (line 510-537) have been included in the manuscript.

(1)

The goals of this study are unclear – is the goal to predict wildfires in the future based on weather conditions, to support climate projections, or to simply estimate the amount of burned area?

The goal of this study is to develop a wildfire burned area prediction model that can be used to quantitatively estimate the contribution of different environmental factors that control wildfires at the grid level. We have stressed the goal of this study in line 69-73. To better represent this goal, we slightly revised the paper title to: "Quantifying the effects of environmental factors on wildfire burned area in South Central US using integrated machine learning techniques"

(2)

A related critique is that the structure of the paper makes it difficult for the reader to follow, there are effectively two methods sections with the results of the first set of methods in the middle.

As suggested, we have moved the validation method (original section 3.1) to Model section (new section 3.2).

(3)

Additionally, the authors never present a figure showing the modeled burned area, which should be the main output of this work and really needs to be emphasized in the main body of the manuscript.

We have moved the original Fig. S1 to the manuscript as Fig. 4 (Fig. R10 below). Fig. R10 shows the maps of monthly mean observed and predicted burned area averaged from 2002-2015 for both fire seasons. In addition to Fig. R10, Fig. 3 and Fig. 5 in the manuscript show the modeled burned area verses the observed burned area at the grid level and at the large-domain level. These results demonstrate that the model has the certain ability in predicting burned area at the grid-scale and at large-domain scale. We have rewritten and reorganized the corresponding paragraphs to emphasize the results in section 4.

[Figure]

Fig. R10. Map of monthly mean observed and predicted burned area averaged from 2002 to 2015 for the (a) winter-spring and (b) summer fire season. (This figure is now Fig. 4. In the revised manuscript)

(4)
The authors have not considered a large body of wildfire research regarding satellite observations-driven modeling which is relevant to this work in the background research. Similar studies involving the effects of climate on total burned area should be noted by the authors, including Andela et al., 2017 and Zubkova et al., 2019.

The focus of this study is on machine-learning based prediction of wildfires and the relative importance of environmental controls of wildfires. Thus, the literature review was mainly focused on this aspect. However, we agreed reviewer's comments and added some references including the references mentioned by the reviewers.

(5)
Additionally, the methods section refers to aspects of the data which are not described until a later section.

As suggested, we have moved the data section (new section 2) before the model section (new section 3).

(6)
This organization is difficult for the reader to follow, and the description of the data used is insufficient, in part because the sources of the input data are not provided.

The sections have been rearranged as Data (section 2), Model (section 3), and Model validation and evaluation (section 4), Contributions of environmental factors to

predicted wildfire burned area (section 5), and Discussion and Conclusion (section 6). To better clarify the data sources, we have included them into the manuscript in section 2.

(7)
The data preprocessing methods are unclear as well – how is a discrete thematic variable like land cover type represented at 0.5-degree resolution?

The land cover type data is represented as 30 m x 30 m pixels with an assigned value to represent a given class of land cover type. We used the nearest neighbor resampling method to regrid the land cover type data onto the lower resolution of 0.5° x 0.5°. The nearest neighbor resampling method is illustrated in Figure R11. This method does not change any value from the original dataset but assigns the value to the new grid according to the value of the grid closest to the center of the new grid. This method was chosen because it is the fastest of the interpolation methods. For instance, compared to the nearest neighbor resampling, the majority resampling is relatively slow and time-consuming, because it is sensitivity to the size of the filter window and thus more experiments are required to determine the filter window. Additionally, since the it does not change the values of the cells, it is widely used for discrete data and it can keep the extreme values that are highly related to large fires (Baboo and Devi, 2010).

We applied the nearest neighbor resampling method to both continuous and discrete thematic variables to the 0.5° resolution. We have added the data preprocessing methods into the Data section (line 109-113).

[Figure]

Fig. R11. Nearest Neighbor resampling technique. In this case, the nearest neighbor resampling is applied to grids with a resolution of 1x1 cell (left) to obtain grids with a resolution of 3x3 cell (right). (adopted from ESRI (2010)).

(8)
How are the translations between quantiles and area being made, given that the area of the grid cell varies with latitude?

The burned area of each grid cell was calculated by interpolating the fire data points into 0.5° grid cell based on their location. Although area of a grid decreases with latitude and higher latitude grids may contain fewer fire data points for interpolation, our interpolated burned area only depends on the magnitude but not the amount of the fire data points in a grid cell. Thus a higher latitude grid could have a large burned area despite its smaller grid area. For instance, we randomly sampled 10% of the grids from

two groups of grids for ten times: grids with latitude ranging from 26.75° to 28.25° (representing larger grids in lower latitude) and grids with latitude ranging from 35.25° to 36.75° (representing smaller grids in higher latitude). As Figure R12 shows, the grids in lower latitude ranging from 26.75° to 28.25° overall have smaller burned area, with the mean log of burned area of 1.28 ± 2.35 ha for the sampled grids, while the grids in higher latitude ranging from 35.25° to 36.75° generally have larger burned area, with the mean log of burned area of 2.18 ± 2.67 ha.

The above analysis supports the argument that a higher latitude grid could have a large burned area despite its smaller grid area. Given that our model can successfully capture burned area in a grid cell across various latitudes and burned area may not be dependent on grid area, our interpolated burned area distributions therefore need not be normalized to cater grid cells with different grid areas. When the quantile of the burned area distribution is predicted by the model, we just need to use the predicted quantile to extract the final predicted burned area from the distribution.

[Figure]

Fig. R12. Probability distribution of burned area for the randomly-selected grids in latitude range of 26.75° to 28.25° (black) and 35.25° to 36.75° (blue).

(9)
Also, the authors say the model predicts burned area at 50 km spatial resolution (with no indication of the map projection used), this is not the same as 0.5 degrees and this discrepancy needs to be resolved.
For this study, the point location of wildfire burned area was grouped into 0.5° grid cell based on their longitude and latitude. To avoid confusion, we have replaced the spatial resolution of '50km x 50 km' with '0.5° x 0.5°' throughout the manuscript.

(10) There are also questions about the fire data used to train the model – are prescribed fires included in the data (by definition, these are not wildfires in most cases)? Is there an estimate of the number of fires which are omitted?
The FPA-FOD fire data that we used excludes prescribed fires except for the prescribed fires that escaped their planned perimeters and became wildfires (Short, 2017). We have

clarified this in the manuscript (line 92-93).

Short (2014) compared FPA FOD data (1992-2011) with two national fire estimates from the US Department of Agriculture Forest Service (USFS) Wildfire Statistics and the National Interagency Coordination Center (NICC), which are available for 1992-1997 and 1998-2011, respectively. They showed that the annual number of fires estimated by FPA FOD is about 30% lower compared to that from the USFS estimation for the period of 1992-1997, as shown below (Fig. R13). The inconsistency of the fire number possibly may be caused by underestimation of small fires, as the fire burned area agrees well with USFS data. Our model will not be able to predict those small fires missing from the FDA-FOD as such information is not in the training dataset. The above discussions have been included in the text (line 93-98).

[Figure]

[Figure]

Fig. R13. Comparison of wildfire (a) numbers and (b) area burned area (hectares) in the US, 1992-2011, from published national estimates (USFS/NICC) and from FPA FOD. (adopted from Short et al. (2014); Fig. 4).

(11) Given the quantile-based approach, what happens if there is a fire or amount of burning which is greater than any in the training dataset (i.e. it would fall out of the range of the training data unless there was a training cell with 100% burned area)?

For a single grid, our four-step model can predict burned area greater than it had before based on its environmental conditions and by learning from other grids. However, random forest or quantile regression forest model cannot predict burned area larger than what it observed before from all the grids. For example, if the largest gridded burned area across the whole domain is 800 ha, the prediction for a single grid would never exceed 800 ha. Even though other methods such as MLR can predict burned area larger

than it observes before, there are some uncertainties in extrapolation (Amatulli et al., 2013; Mckenzie et al., 1996). We have included the above discussion in the manuscript (line 513-520).

(12)
Is the length of the training dataset long enough to capture all variability in fire activity as it relates to climatic conditions?
In 10-fold cross validation, the training dataset contains 16277 samples (It is derived from the total data length of 18085/10*9=~16277) for each fold for the winter-spring fire season. Assuming fire-climate relationships are unique for each individual grid, the large sample size is enough to capture all the variability in fire activity and its response to recent decadal climate. We have included this statement into the manuscript in line 429-432.

(13)
Why were remote sensing-derived datasets not considered?
We included satellite-derived monthly mean Leaf Area Index (LAI) obtained from MODIS instrument. In terms of fire data, since our focus is on wildfires and remote-sensing dataset does not separate prescribed fires from wildfires, we used FPA-FOD fire data instead of satellite-derived fire data such as GFED4 or FINN.

(14)
An important aspect of fire regimes which was not adequately considered in the manuscript is the role of human activity in the fire regime, especially in the United States where humans play an active role in the fire regime through suppression, ignitions, fuel load management, and landscape fragmentation in addition to being the source of ignition of approximately 85% of fires (according to the US Forest Service). These effects vary as function of not only population density, but sociopolitical norms which can vary from state to state. Recent papers such as the Andela et al. 2017 paper claim human activity is the major control on fire activity, and as such it cannot be ignored in a study region where the fire regime is likely human-driven. While the datasets describing human activity are certainly far from perfect, it is not possible to describe fire activity in a human-driven fire regime without considering human influences.
The focus of the paper is to quantify how environmental factors control wildfires in the study region under the present-day human management practices and human activities. Thus, we only included population density data of year 2010 to represent present-day human influence on wildfire activity. We acknowledge that population density is a rough estimate of effect of human activity on fires. We have included this statement into the manuscript in line 523-528.

(15)
Finally, there needs to be more effort in describing the expected impact of the work and the limitations of the method. For example, the abstract mentions that the work can be

used to assess fire management strategies but provides no details on how or why.

To clarify, the developed model aims to provide a broader impact on the community by accessing the quantitative contributions of the environmental controls of wildfires. An improved understanding of relative importance of the factors on wildfires would be useful for future fire prediction, fire management, as well as the linkage between wildfires and climate change. We removed the specific use of the model for fire management in the abstract and have restated the expected impact of the work and added the limitation of the method in the manuscript (line 510-537 and 558-567).

(16)

The quality of the input data is not discussed, which will propagate errors through the model, as well as the serial structure of the integrated model itself. At present, the manuscript is too focused on the machine learning exercise rather than on the scientific value of the work.

The discussion about the quality of the input data is added into the manuscript (line 520-523). Also, we have rewrote the manuscript to emphasize the scientific value of this work.

Specific Comments:

1. L75: Why were other months of the year excluded?

The total burned area of the two seasons accounts for 76% of the total annual burned area. As Fig. R14 shows, there are two peak seasons in South Central US: January to April and July to September. The dominance of wildfire occurrences in these months implies natural environmental conditions in these months are most conducive for wildfires. While wildfires do occur outside the fire seasons, their lower frequency implies that non-natural factors (e.g. human actions) can be relatively more important. As our study did not focus on human factors, we chose to exclude other months of the year. We have included the reason (line 74-79) and the Fig. R14 into the manuscript.

[Figure]

Fig. R14. Total burned area by month over South Central US. (This figure is now Fig. S1. In the revised manuscript)

2. L81: "Uneven data" is used throughout the paper but is not defined. Does this refer to unevenness spatially, temporally, or both?

It refers to both. The uneven distribution of burned area is defined as the situation where the number of grids with large burned areas is much smaller than the number of grids with small or zero burned areas. This situation exists for a single grid (temporal unevenness) and for all the grids within a given time period (spatially). The definition of unevenly-distributed burned area data has been included in the manuscript (line 57-59 and 182-183). An example of the uneven distribution of gridded burned area in winter-spring fire season is shown in Fig. R15. The Fig. R15 is included into the supplement to support the example of our case.

[Figure]

Fig. R15. Histogram of gridded burned area for the winter-spring fire season. (This figure is now Fig. S2a. In the revised manuscript)

3. L92-95: Given that the model compares the output to the quantile ranges, is it capable of estimating an amount of burning greater than has been observed in the training data?
See the response to general comment #11 above.

4. L155: Is there any concern about propagation of error through the model? What is the benefit of running three models in serial rather than one model alone or several models in an ensemble?
See the response to comment 6 for the reviewer 1, as shown below:
The reviewer is correct that biases from one step could be propagated to the subsequent steps, for example when the burned grids are predicted not to burn or when the unburned grids are predicted to burn. For the first case, when the burned grids are incorrectly predicted not to burn, the low bias is introduced because the burned grids would not proceed to step 3. For the second case, inclusion of unburned grids in step 3 may introduce a positive bias. We have included the discussions of the error propagation in section 6 (line 511-513).

To demonstrate our four-step model can achieve a higher accuracy and alleviate the issue of uneven-distributed dataset, we compare the prediction performance using

random forest alone with that of the four-step model developed in this study, as shown in the Table R3 below:

Table R3. Comparison of MAE and skewness between the RF model and the developed model (The information of this table is now included into the Table S2 in the revised manuscript)

| Model | RF alone | Model developed in this study |
|---|---|---|
| MAE (winter-spring) | 1.34 | 1.13 |
| Skewness (winter-spring) | 37.40 (burned area) | 0.70 (quantiles) |
| | | |
| MAE (summer) | 0.70 | 0.57 |
| Skewness (summer) | 33.83 (burned area) | 0.96 (quantiles) |

Skewness is a measure of the asymmetry of the probability distribution of a random variable about its mean. The skewness of a random variable X is the third standardized moment $\widetilde{\mu_3}$, defined as:

$$\widetilde{\mu_3} = E\left[\left(\frac{X-\mu}{\sigma}\right)^3\right] = \frac{\mu_3}{\sigma^3} = \frac{E[(X-\mu)^3]}{(E[(X-\mu)^2])^{3/2}} = \frac{\kappa_3}{\kappa_2^{3/2}}$$

where $\mu$ is the mean, $\sigma$ is the standard deviation, E is the expectation operator, $\mu_3$ is the third central moment, and $\kappa_t$ are the $t$-th cumulants. If skewness is less than -1 or greater than +1, the distribution is highly skewed. If skewness is between -1 and -0.5 or between +0.5 and +1, the distribution is moderately skewed. If skewness is between -0.5 and 0.5, the distribution is approximately symmetric. The positive value indicates that the tail is on the right side of the distribution while negative value indicates that the tail is on the left.

Our model has a lower MAE, by 15% and 19% for the winter-spring and summer fire season, respectively, compared to the single RF model. The distribution of the quantiles in the developed model is more uniform than the distribution of the burned area, as shown in Figure R6 below and the skewness. We have added a discussion of this issue in the text (line 355-365) and the Table R3 in the supplementary information. The information of skewness calculation has been included in the supplementary information. Note that we have replaced 'quantile' with 'percentile' in the manuscript to better clarify the idea of quantiles.

[Figure]

Fig. R6. (a) Histogram of burned area (b) Histogram of the percentile groups of burned area for the winter-spring fire season. (This figure is now Fig. S2. In the revised manuscript)

5. L164: Is there an estimate of the number of fires missed? Small fires constitute most of the fires by number, even though they add up to relatively little burned area (e.g. Malamud, Millington, and Perry 2005). It is noted that the dataset omits most small fires occurring on private land – these are not generally wildfires and such fires should be omitted anyways if the study is about wildfires.
See the response to general comment #(10) above.

6. L194-195: I don't think the climatic variables can be considered as fixed, especially since the assumption in later parts of the paper surround climate change scenarios which means their values do vary through time.
Climatic variables that are considered as fixed include only the mean and standard deviations of monthly meteorology over the past 22-years (1979-2000), because they do not vary by time over our study period (2002-2015) but characterize the spatial patterns of wildfire occurrence and intensity. The variables of climate anomaly are classified as climate variables (as opposed to fixed variables) since they are defined as the difference between monthly mean and the long-term average over 1979-2000 and their values vary by time. We have clarified this in the text (line 149-150 and line 435-440).

7. L209: Is any consideration given to preventing overfitting due the correlation between variables? For example, ecoregions and landcover types are likely to be related to one another.
Yes, we considered the collinearity of the variables when we designed the model. Thus, we chose logistic model and random forest model which work reasonably well under moderate collinearity (correlation coefficient < 0.7) (Dormann et al., 2013). We have added the concern of collinearity between variables into the text in line 213-215.

Although ecoregions and landcover types are likely to relate to one another, ecoregions represent large-scale areas comprised of similar biotic and abiotic phenomena while

land cover types are able to provide more detailed land information within one ecoregion. For example, in the temperate prairies (one ecoregion of our study domain), it has complex land types including pasture, woody wetlands, evergreen forest, and cultivated crops. Inclusion of these two variables allows us to capture fire responses to large-scale ecoregions and small-scale land types.

8. L232: Please clarify the phrase "horizontal scale of around 700 x 700 km2" – the use of horizontal scale implies a one-dimensional unit (length) which does not match the unit specified. Also, as a suggestion, 700 x 700 km2 seems ambiguous and could be more clearly represented as "700 km x 700 km" or "490,000 km2"

Good point. To avoid confusion, we have changed 'horizontal scale of around 700 x 700 km2' to 'spatial scale of around 700 km x 700 km'. The similar changes were also made for the Table S1.

9. L252: SUS is never defined

We have defined 'SUS' as 'southern US' for clarification.

10. L268-270: One could argue that the model is in fact "hardwired" (editorially, the term is jargon and should be replaced) to the geographical features of the study domain – geography deals with the human components of space and time as well as the physical components. The tendency of the human population to ignite or suppress fires as a result relationship to sociopolitical factors (like local regulations) will influence the fire regime in ways which will not be captured by climatic variables and will change from location to location.

We agreed with reviewer's point but the main focus of this paper is on how the environmental factors control wildfires in South Central US under the present-day human management practices and human activities. Therefore, the geographical features in the manuscript refer to coordinate variables such as longitude and latitude. To clarify this, we have replaced the term 'hardwired' and rewrote the corresponding paragraph in the manuscript (line 348-351).

11. L286: Why were 14 variables chosen? This seems like an arbitrary cutoff, especially given the large number of variables which went into the model.

We chose the top 14 variables because they represent the top quarter (25%) of the selected predictor variables. %IncMSE represents the change of mean square error with and without permuting variables. A larger %IncMSE value represents a higher variable importance. To see the sensitivity of the importance to the variable rank, we calculated the ratio of %IncMSE at variable ranked as $X^{th}$ percentile (~top Y) to the %IncMSE at variable ranked as top (Y+1). Larger ratio means larger drop-off of the %IncMSE between topY and top(Y+1), which indicates notable decrease of variable importance at the cut-off point (top Y+1). We compared the ratio at several cutoff points: $25^{th}$ percentile, $50^{th}$ percentile, and $75^{th}$ percentile:

Table R6. The ratio of %IncMSE at variable ranked as $X^{th}$ percentile ($Y^{th}$) to the %IncMSE at variable ranked as $(Y+1)^{th}$ for the three selected percentiles

|  | 25% (Y=14) | 50% (Y=29) | 75% (Y=43) |
|---|---|---|---|
| Winter-spring fire season | 1.21 | 0.88 | 1.00 |
| Summer fire season | 1.06 | 1.01 | 1.00 |

As the table shows, 25-percentile cut-off point has largest ratio, indicating a large drop of variable importance at the variable ranked $15^{th}$ and the top 14 variables have significantly larger importance. Thus, the top 25% variables (the top 14 variables) were chosen to be further discussed. The reasons of choosing the top 14 variables and associated discussions have been included in the manuscript (line 380-382). Table R6 has been added into the supplementary information as Table S4.

12. L306: The fuel-related variables are among the least important presented in Figure 5 – how can the conclusion be drawn that fuel abundance is what determines the amount of burned area?
Although fuel-related variables are among the least important in the top 14 variables, they are the fifth and sixth most important variables when excluding the fixed variables. Our conclusion was mainly based on the importance of time-varying variables. Therefore, burned area in the winter-spring fire season is mainly controlled by RH anomaly that directly affects fuel moisture. Besides that, the antecedent fuel abundance and pre-fire-season drought conditions together determines the amount of dry fuel in the winter-spring fire season. To clarify this, we have rewritten the corresponding paragraph in line 407-410.

13. Table 1: The resolution of the data is presented, but it's not clear how the data are being re-gridded to the working resolution - if the fire analysis is being done at 0.5 degree and the climate data is at 32 km resolution then there are < 4 cells per burned area data point and the way which those 4 cells are represented has significant consequences.
See the response to general comment #(7) above.

**References:**
Amatulli, G., Camia, A. and San-Miguel-Ayanz, J.: Estimating future burned areas under changing climate in the EU-Mediterranean countries, Science of The Total Environment, 450–451, 209–222, doi:10.1016/j.scitotenv.2013.02.014, 2013.

Baboo, S. and Devi, R.: An Analysis of Different Resampling Methods in Coimbatore, District, Global Journal of Computer Science and Technology, 10(15), 61–66, 2010.

Dormann, C. F., Elith, J., Bacher, S., Buchmann, C., Carl, G., Carré, G., Marquéz, J. R. G., Gruber, B., Lafourcade, B., Leitão, P. J., Münkemüller, T., McClean, C., Osborne,

P. E., Reineking, B., Schröder, B., Skidmore, A. K., Zurell, D. and Lautenbach, S.: Collinearity: a review of methods to deal with it and a simulation study evaluating their performance, Ecography, 36(1), 27–46, doi:10.1111/j.1600-0587.2012.07348.x, 2013.

ESRI: Manual ArcGIS [software GIS] Version 10, Environmental System Research Institute, Inc., 2010.

Mckenzie, D., Peterson, D. L. and Alvarado, E.: Extrapolation Problems in Modeling Fire Effects at Large Spatial Scales: a Review, Int. J. Wildland Fire, 6(4), 165–176, doi:10.1071/wf9960165, 1996.

Short, K. C.: A spatial database of wildfires in the United States, 1992-2011, Earth System Science Data, 6, 1–27, doi:10.5194/essd-6-1-2014, 2014.

Short, K. C.: Spatial wildfire occurrence data for the United States, 1992-2015, Forest Service Research Data Archive (4th Edition), doi:10.2737/RDS-2013-0009.4, 2017.

---

## Author Response (AR2)

**Response to Reviews**

We thank the two reviewers for their constructive comments to improve the manuscript. Their comments are reproduced below with our responses in blue. The corresponding changes in the manuscript are highlighted in blue.

**Reviewer #1**

The revised version has been improved largely. The remaining comments are about the parameters of XGboost method and the reliability of the RF importance index.

1. Could you describe the parameters of the XGboost method, such as the number of trees, the maximum tree depth, et. al.

Response: The hyperparameters of XGboost were tuned by a grid search with 10-fold cross-validation to find the best model based on MAE. Table R1 shows the optimum value of each hyperparameter for the winter-spring and summer fire season model.

**Table R1.** The selected XGboost hyperparameters for the winter-spring and summer fire season

|  | eta | max_depth | gamma | subsample | colsample_bytree | min_child_weight |
|---|---|---|---|---|---|---|
| Winter-spring | 0.01 | 10 | 3 | 0.75 | 0.7 | 1 |
| Summer | 0.05 | 8 | 3 | 1 | 0.6 | 1 |

We have included the parameters of the XGboost model into the manuscript and a brief introduction about the algorithm into the supplementary (Table S3).

2. Although you show the RF importance is stable under cross-validation, only the model with high accuracy presents true physical relationships. Because of some limitation factors, such as the number of samples and the strong non-linear problem, the machine learning method may be stable under a low accuracy, but it can not prove that the importance index is reliable. I think the authors can verify it by physical knowledge. It is also because the machine learning method is black box, and we want to get knowledge by combining physics and machine learning.

Response: the reviewer's point is well taken. Although we cannot directly prove the ranks of the variables are correct (e.g. the variable that is ranked first is the most important variable than all the other ones), the selected top 14 variables all have physical linkages to burned area that have been discussed in prior studies. For example, RH anomaly and temperature anomaly are the only climate anomaly variables in the top 14 variables for the summer fire season. The physical reason behind their importance is that higher temperature coupled with lower relative humidity in the summer can cause drier fuel and this condition is favorable for fires to start, spread, and burn more intensely (Williams et al., 2013; Holden et al., 2018). In the manuscript, we demonstrate the stronger correlation between RH and temperature anomaly in summer than in winter-spring (line 426-431). We have included the discussions of the top 14 variables and verified the selection by physical knowledge and prior studies in section 5.1. We have also added the uncertainties in the rank of RF importance in the manuscript (line 495-497).

**Reviewer #2**

This manuscript integrated multiple machine learning algorithms and developed a prediction model at the resolution of 0.5ox0.5o to predict monthly wildfire burned area over the South Central United States during 2002-2015. The relative importance of the environmental drivers is also identified. The paper was well structured and presented. It can be accepted for publication if the following questions can be addressed.

1. In Figure 6, both rhum_anomally and rhum are selected as important predictors for the winter-spring fire season. What is the difference between them and are they highly correlated? Also, please show the correlation coefficients between the variables in Figure 6.

Response: For a given grid, the rhum is the mean RH for a month and rhum_anomaly is the difference between rhum and the long-term climatological (1979-2000) mean RH of the same grid and same month. In other words, the rhum is the actual RH which can vary by location and season, while rhum_anomaly measures the departure of rhum from its long-term average due to climate change and/or climate variability. For the study domain and time period, the correlation between rhum_anomaly and rhum is 0.66 (Fig R1). Although they have a moderate correlation, their values have different meanings and both of them are included in the model. For example, for grids with rhum of ~70%, rhum_anomaly can range from -11.16% to 15.35%. For the same rhum value of ~70%, positive rhum_anomaly indicates a relatively wetter condition and negative rhum_anomaly a relatively dryer condition compared to their long-term condition in the past. We have included the discussion of the differences between these two variables in the manuscript (line 416-423).

[Figure]

**Fig. R1.** Comparison between RH (rhum) and RH_anomaly (rhum_anomaly) for the winter-spring fire season.

The correlation coefficients between the predictor variables are shown in Figure R2 below.

[Figure]

**Fig. R2.** The correlation plot of the top 14 variables for the (a) winter-spring and (b) summer fire season. (This figure is now Fig. S9. In the revised manuscript)

The correlation plot shows that most of the important variables have weak to moderate correlation ($r < |0.7|$) between each other. The exceptions are for the fixed-climate variables (e.g. asum_avg vs. apcp_avg and temp_sd vs. tmax_sd) and the antecedent variables (e.g. SPEI_mean4m and SPEI_mean5m) for both fire seasons. This is expected because the long-term mean or standard deviation of the same types of meteorology do not change by time and the average of antecedent drought conditions (SPEI) may not vary a lot from including or excluding a single month. Although there is collinearity between the predictor variables, the logistic model and RF model we used in this study are relatively insensitive to collinearity. The discussion about the sensitivity of the threshold of collinearity can be seen the response to comment #13 for the reviewer 3. We have included the above discussion into the manuscript (line 469-481). We also noted that the threshold of 0.7 in the original manuscript is only applied to the pairs of the time-varying predictor variables, except for the averaged SPEI for the pre-fire season (line 225-229).

2. In section 4, the authors compare the performance of the prediction models using MLR to predict burned area in Europe, Western US, etc. Please add several figures to show the performance of MLR model in the prediction of burned area in South central US using the same variables in Table 1 as a comparison of the model using 4-step machine learning.
Response: We compared our model results with MLR and combined it with the results of some other machine learning methods (the parameters of each model have been optimized):

**Table R2.** Comparison of MAE and skewness between the developed model, RF model, XGBoost model, and MLR model (This table is now Table. S2. In the revised manuscript)

| Metrics | Model developed in this study | RF alone | XGBoost alone | MLR alone |
|---|---|---|---|---|
| MAE (winter-spring) | 1.13 | 1.34 | 1.26 | 1.44 |
| Skewness (winter-spring) | 0.70 (quantiles) | 37.40 (burned area) | 37.40 (burned area) | 37.40 (burned area) |

|  |  |  |  |  |
|---|---|---|---|---|
| MAE (summer) | 0.57 | 0.70 | 0.67 | 0.76 |
| Skewness (summer) | 0.96 (quantiles) | 33.83 (burned area) | 33.83 (burned area) | 33.83 (burned area) |

Our four-step model has a lower MAE, which is 27% and 33% lower than the MLR model for the winter-spring and summer fire season, respectively. The developed model presents a better performance in predicting burned area than MLR model and the other two models (Fig R3). The distribution of MAE from 10-fold cross validation shows that our four-step model has a smaller median MAE but larger range of MAE compared to other models. We have included the results of MLR model into Table S2 and Figure R3 as Figure S6 as supplementary (line 376-382).

[Figure]

**Fig. R3.** Box plots of MAE from 10-fold-cross validation and different methods for (a) winter-spring and (b) summer fire season. (This figure is now Fig. S6. In the revised manuscript)

**Reviewer #3**

This manuscript proposes a new method for modeling wildfire burned area that includes a combination of random forest, quantile regression and logistic regression which should help with the issue of unevenly distributed burned area data. This technique was applied for modeling the amount of burned area in the South Central United States during two fire seasons. The author concluded that antecedent climate conditions, in particular relative humidity, is the main driver of the amount of burned area in the winter-spring season while concurrent weather conditions drive fire activity during summer months. However, this proposed method has several limitations that some might argue are substantial and require more attention than being listed at the end of the manuscript. Additionally, the lack of information regarding how this technique can be applied to different regions with longer fire return interval and/or different datasets makes the proposed method more questionable. Moreover, purposely excluding all non-natural fires and out of more than 30 variables selecting only one not related to climate (population density), makes the main conclusion that climate is the main driver of fire activity biased especially considering that most of the fires in the US have an anthropogenic origin. All things considered, I would recommend a major revision.

Response: We appreciate the comments from the reviewer. We have included the discussion about fire return interval and future application of our model to different regions (see our response #21). In terms of the selection of variables, besides climate variables, we also include fuel variables (LAI,

conv_LAI, and soil moisture), weather variables (fire-season monthly mean meteorology), and geospatial variables (land types, ecoregion types, and population). These variables are related to burned area and have been widely discussed in prior studies. When the variables were assigned into four groups of different environmental controls, we scaled the contribution of each group by the number of variables when quantifying the contribution of each group. Thus, the calculated relative importance is not affected by the number of variables included in the group (see our response #26 and #27). We have included the above discussions in the revised manuscript.

1. P1L20 and L22: I would recommend replacing "the magnitude of total burned area" to "the amount of total burned area"
We have changed the term to "the amount of total burned area" (line 22).

2. P1L27: You cannot say "many regions of the world" have experienced an increase in fire activity while only citing papers from the US. You should either add citations from different regions of the world or only write about the US.
We have added more references for different regions of the world (line 28-29).

3. P2L37-38: Please reword this sentence. "…complex interplay …can change by spatial scale…" does not sound correct.
We have changed the sentence to "… can vary depending on spatial scale…" (line 38).

4. P3L73: Can you explain the choice of the study area? In the title and all over the paper you claim to model fire activity over South Central US while your actual study area is not a geographic region or a set of ecoregions. It is an arbitrary rectangular. How is that a "vegetation rich" part of South Central US? Is it based on a particular vegetation map? If so, what was the threshold?
Additionally, it will be beneficial to include a paragraph or to describe the study area after the introduction. To explain in more detail why it is important to model fire activity in that particular part of the US since most studies concentrate on California and Pacific Northwest, regions more fire-prone comparing to South Central US. Also a detailed description of its fire regime, vegetation and climate is necessary.
The South Central US domain was selected based on several reasons. First, this region is composed of similar vegetations which are plains and oak-hickory forests. Second, the domain was selected as a rectangular and excluded western Texas because the majority of wildfires occurred over the eastern Texas and lots of them were over populated area, as shown in Fig R4. Last but not the least, most prior studies focused on the western US, while the South Central US had experienced periodically large wildfires in recent years and is projected to have the highest risk of wildfires in 2031-2050 across the continental US (Fig R5).

[Figure]

**Fig. R4.** Maps of wildfires during 2005-2011 over Texas. Wildfires occurred within two miles of a community are colored by red and wildfires did not occur within two miles of a community are colored by black (adopted from Jones et al. (2013)).

[Figure]

**Fig. R5.** Changes in wildfire risk relative to the baseline (historical average for 1991-1997) with the future climate (2031-2050) projected by (a) HadCM3 model under the RCP 4.5 scenario, (b) NOAA-GFDL model under the RCP 4.5 scenario, (c) HadCM3 model under RCP 8.5 scenario, (d) NOAA-GFDL model under RCP 8.5 scenario (adopted from Ann et al. (2015)).

The selection of the vegetation-rich part of South Central US was based on the tree canopy map from the National Land Cover Database (NLCD) 2016 (Fig R6). Each pixel represents the percentage of tree canopy and darker-green pixels show larger percentage of tree canopy. As Fig R6 shows, the study domain (eastern Texas, Oklahoma, Louisiana, and Arkansas) has a relatively higher percentage of tree canopy.

[Figure]

**Fig. R6.** Tree canopy map of the continental US from the National Land Cover Database (NLCD) 2016. The red box denotes the selected South Central US domain.

The vegetations over the South Central US are composed of plains and oak-hickory forests. More than 70% of the fires over this region can cause less than 25% upper layer replacement but burns 5% of more of the area, with maximum and minimum interval year of 300 years and 1 year (Barrett et al., 2010). Typical wildfire seasons over this region are winter-spring (Jan-Apr) and summer (Jul-Sep) (Fig S1). The winter-spring wildfire season is characterized by the large circulation patterns associated with a low-pressure center usually producing drier-than-normal and strong winds (Heilman et al., 1998; Jones et al., 2013). Wildfires in the summer are mostly driven by abundance of dry or dead vegetations during dry season (Jones et al., 2013). We have included the above descriptions about vegetation, fire regimes, and fire characteristics into the introduction (line 65-72 and line 77-82).

5. P3L94: Here you provide the accuracy of FPA-FOD for the period 1992-1997. How is this relevant to your study when your study period is 2002-2015? Your study period is covered by the MODIS fire product. Why not to use global fire product instead? Usually, the benefit of using regional data is its accuracy; however, here small fires which I assume are the majority in South Central US are not included. Additionally, most places in the world do not have datasets similar to FPA-FOD. If the results of your proposed method are sensitive to the choice of the dataset does it mean that this method can be only used within the US?

Although the evaluation of FPA-FOD for the period of 1992-1997 is not directly relevant to our study period of 2002-2015, it still provides the caveat or uncertainty that we might need to consider when using this dataset.

Our focus is on wildfires while the satellite-derived global fire products do not separate prescribed fires from wildfires. Satellite fire products can also miss small fires that are below clouds or occur outside the satellite overpass time. Fig R7 and Table R3 below show the fire statistics between the FPA-FOD and the Fire INventory from NCAR (FINN) that are based on MODIS fire products over the study domain during 2002-2015. Compared to FINN, FPA-FOD records larger numbers of fires and includes more small and large fires. The majority of fires recorded in FPA-FOD are small, as shown by the median burned area. Fewer fires are observed by satellites, but the fires that can be detected by satellites are usually larger than a certain size. Therefore, FINN shows a larger

total burned area and mean burned area. Although some small fires might be omitted in FPA-FOD data, FPA-FOD dataset provides more information about small wildfires compared to FINN.

Our proposed model aims to address the issue of unevenly-distributed fire data. Regardless of the fire data source, the proposed model can be applied to other regions.

**Table R3.** Comparison of fire statistics for the FPA-FOD and FINN fire data during 2002-2015

|  | Number of fires observed | Total burned area (km$^2$) | Mean burned area (km$^2$) | Maximum/Minimum burned area (km$^2$) | Median burned area (km$^2$) |
|---|---|---|---|---|---|
| FPA-FOD | 116540 | 13610.58 | 0.117 | 512.8/0.00004 | 0.004 |
| FINN | 67553 | 52719.55 | 0.780 | 0.0375/1.00 | 0.75 |

[Figure]

**Fig. R7.** Distributions of fire size from FPA-FOD and FINN fire data.

6.Table1: It is not clear that the source of most of the weather/climate variables is NARR, neither their resolution.
We have revised the Table 1 to make the source and resolution of weather/climate variables clear.

7. P4L116: You need to cite every dataset that was used in this paper.
We have added the references for the dataset into the paper.

8. P4L122: Citation is needed.
We have included the corresponding references into the paper (line 127).

9. P5L127: "Situations" is not a good choice of words.
We have rewritten the sentences (line 133).

10. P5L155: Citation is needed.
We have included the corresponding references into the paper (line 161-162).

11. P6L171: What exactly is "ecoregion type"?
We have added additional descriptions about ecoregion type and its differences between land cover

types in line 180-185.

12. P6L174: What is "EPA"?
We have defined 'EPA' as 'United States Environmental Protection Agency (US EPA)' for clarification (line 183).

13. P7L215: I would argue that 0.7 is a very strong correlation. Lower correlation to 0.5 will help to reduce the number of predictor variables which is way too many and most of them represent similar processes.
We conducted a sensitivity test where the model uses predictor variables that have lower degrees of collinearity (i.e., the correlation coefficient between each pair of the predictor variables is less than 0.5 as suggested by the reviewer. Table R4 shows the predictor variables included in the sensitivity test in comparison to the original model. Generally, removing the predictors that have a higher degree of collinearity causes larger biases in classifying burned grids and predicting extremely-large fires (Table R4). The overall MAE and RMSE also slightly degrade in the sensitivity test. The better performance of the original model makes us to decide to use the threshold of 0.7 and include all the variables in the model. Although some variables may have a moderate correlation, their values have different meanings and thus can be interpreted differently, as stated above for the case of rhum and rhum_anomaly. We have included the above discussion into the manuscript and Table R4 in the supplementary (line 475-481).

**Table R4.** Comparison of accuracy, AUC, F-1 score, MAE, RMSE, and MAE of large burned area between the model with the chosen set of predictors and the model with the predictors that have lower degrees of collinearity ($r < |0.5|$) (This table is now Table. S7. In the revised manuscript)

| Model | Model with the chosen set of variables | Model with variables that have low degrees of collinearity |
|---|---|---|
| Number of predictor variables | 58 | 33 |
| Accuracy (winter-spring) | 0.74 | 0.71 |
| AUC (winter-spring) | 0.82 | 0.78 |
| F-1 (winter-spring) | 0.79 | 0.77 |
| MAE (log of area; winter-spring) | 1.37 | 1.43 |
| RMSE (log of area; winter-spring) | 2.03 | 2.06 |
| MAE of large burned area[†] (log of area; winter-spring) | 2.32 | 2.57 |
| | | |
| Number of predictor variables | 57 | 31 |
| Accuracy (summer) | 0.74 | 0.72 |
| AUC (summer) | 0.83 | 0.80 |
| F-1 (summer) | 0.77 | 0.75 |
| MAE (log of area; summer) | 1.17 | 1.20 |
| RMSE (log of area; summer) | 1.87 | 1.88 |
| MAE of large burned area[†] (log of area; summer) | 2.25 | 2.31 |

† Large burned area here is defined as the burned area larger than 90th percentile.

**Table R5.** Variables selected for the sensitivity test for the two fire seasons

| Season (number of variables) | Weather | Fuel | Climate | Fix-geospatial |
|---|---|---|---|---|
| Winter-spring (33) | Temp, RH, U, SPEI, LargestConsec, apcp_mean1m, SPEI_mean4m | LAI, soil moisture, | Temp_anomaly, asum_anomaly | Land types, ecoregion types, population |
| | | | | |
| Summer (31) | Temp, U, LargestConsec, apcp_mean1m, temp_mean1m, SPEI_mean1m | Soil moisture | RH anomaly | Temp_sd, land types, ecoregion types, population |

14. P8L252: "Larger" should be replaced by "greater".
We have replaced "larger" with "greater" (line 267).

15. P10L306-330: While I understand that comparison to other work is important, I am not sure that selected studies are comparable. I would argue that there are a lot of studies that modeled the amount of burned area with R2 higher than 0.4. I agree with the authors that funding a study that targeted South Central US is challenging since it is not a region particularly interesting in terms of fire activity. I would suggest applying this method to other regions in the US to provide evidence that this method can be used outside of the study area which has very little fire activity.
The study area of South Central US is actually prone to wildfires and has experienced large fires in recent decades (e.g. the 2011 Texas wildfires). There is no prior study predicting wildfires in this region and that's what motivated our work. The selected studies for comparison were chosen based on the prediction resolution (spatial and temporal) and the approaches (i.e. excluding unburn grids or not). Many studies have modeled the burned area with $R^2$ larger than 0.4, but their temporal and spatial resolution are coarser than ours (Table S1). It is easily understood that the prediction accuracy of wildfire burned areas will increase as the spatial It is easily understood that the prediction accuracy of wildfire burned areas will increase as the spatial and temporal resolution of the prediction model decreases, so the direct comparison without considering the approach and the resolution of different models is not applicable.

The machine-learning based prediction framework can be readily adopted to other regions, but the training of the models are data driven and thus data sensitive. Some selected predictor variables, such as the averages of LAI and sum of neighboring LAI for the months *t-1 to t-6*, were specific to the study domain. Additionally, the quantiles were chosen to include the whole conditional distribution that is specific to the study domain (Fig S3). It would require different sets of predictor variables and selections of quantiles to apply this method to other regions in the US. Considering the focus of this study and the length of the manuscript, we can only include the results of the South Central US. We will leave the application to different regions as a future direction.

16. TableS3: Why the results are presented for 2011 both seasons together, 2014 winter-spring, 2008 summer season? That is confusing. Can you include the results for each year, each season.
The reason why we presented the results of 2011, 2014, and 2008 is to highlight the model ability of predicting wildfires in both the peak seasons and normal seasons and those peak/normal seasons may not occur in the same year. We have included the results for each year and season in terms of $R^2$, RMSE, and MAE in the supplementary (Table S5), as shown below.

**Table R6.** Model performance at grid level for each year (including and excluding the misclassified grids) (This table is now Table. S5. In the revised manuscript)

| | 2002 | 2003 | 2004 | 2005 | 2006 | 2007 | 2008 | 2009 | 2010 | 2011 | 2012 | 2013 | 2014 | 2015 |
|---|---|---|---|---|---|---|---|---|---|---|---|---|---|---|
| **Winter-spring (excluding misclassified grids)** | | | | | | | | | | | | | | |
| $R^2$ | 0.70 | 0.76 | 0.77 | 0.64 | 0.38 | 0.52 | 0.51 | 0.38 | 0.54 | 0.36 | 0.56 | 0.55 | 0.41 | 0.61 |
| MAE (km²) | 0.20 | 0.46 | 0.34 | 0.34 | 5.63 | 0.37 | 2.10 | 3.12 | 0.38 | 3.17 | 0.32 | 0.27 | 0.83 | 0.34 |
| RMSE (km²) | 2.02 | 2.30 | 1.50 | 1.91 | 14.61 | 1.64 | 11.18 | 14.21 | 1.81 | 23.34 | 2.48 | 0.91 | 6.04 | 1.92 |
| **Summer (excluding misclassified grids)** | | | | | | | | | | | | | | |
| $R^2$ | 0.40 | 0.46 | 0.60 | 0.62 | 0.59 | 0.31 | 0.59 | 0.43 | 0.47 | 0.40 | 0.49 | 0.56 | 0.37 | 0.41 |
| MAE (km²) | 0.08 | 0.17 | 0.09 | 0.52 | 0.93 | 0.02 | 0.42 | 0.58 | 0.33 | 3.71 | 1.20 | 0.32 | 0.15 | 0.68 |
| RMSE (km²) | 0.32 | 1.88 | 0.92 | 2.08 | 2.48 | 0.09 | 1.42 | 6.37 | 0.95 | 12.08 | 9.23 | 1.81 | 1.66 | 4.35 |
| **Winter-spring (including misclassified grids)** | | | | | | | | | | | | | | |
| $R^2$ | 0.40 | 0.52 | 0.49 | 0.39 | 0.35 | 0.27 | 0.28 | 0.29 | 0.31 | 0.23 | 0.35 | 0.28 | 0.30 | 0.40 |
| MAE (km²) | 0.18 | 0.35 | 0.26 | 0.31 | 4.84 | 0.33 | 1.92 | 2.83 | 0.311 | 2.78 | 0.25 | 0.24 | 1.05 | 0.35 |
| RMSE (km²) | 1.70 | 1.95 | 1.23 | 1.78 | 13.42 | 1.59 | 10.26 | 13.24 | 1.55 | 21.12 | 2.10 | 0.81 | 5.19 | 3.21 |
| **Summer (including misclassified grids)** | | | | | | | | | | | | | | |
| $R^2$ | 0.28 | 0.10 | 0.28 | 0.33 | 0.45 | 0.10 | 0.42 | 0.29 | 0.31 | 0.39 | 0.32 | 0.40 | 0.20 | 0.26 |
| MAE (km²) | 0.09 | 0.18 | 0.11 | 0.42 | 0.76 | 0.05 | 0.38 | 0.48 | 0.31 | 3.09 | 1.08 | 0.27 | 0.19 | 0.57 |
| RMSE (km²) | 0.67 | 1.75 | 0.95 | 1.79 | 2.20 | 0.25 | 1.58 | 5.50 | 0.96 | 10.85 | 8.35 | 1.63 | 2.06 | 3.85 |

17. P11L338: The results show that the proposed model performs worse if the amount of burned area is outside of the norm in terms of MAE and RMSE. I would say significantly worse. Doesn't it contradict your conclusion that the proposed technique can be used to model future fire activity due to the changing climate? If the model cannot predict unusually high fire activity which as the authors mentioned in the introduction is recently observed in the US than how can it be used for future predictions?
Even though larger MAE and RMSE were shown in 2011 (peak year), our model predicted significantly larger mean gridded burned area for the peak months, as shown in Table R7 below. Additionally, considering the large-scale domain, our model is able to capture the interannual variability of wildfires, as shown in Figure 5. These demonstrate the model ability of predicting large burned area thus it can be used for future prediction. We have highlighted the ability of our model in reproducing the variability of wildfires in the manuscript (line 354-355 and line 397-399).

**Table R7.** Comparison of accuracy, recall, precision, and F1-score between the logistic regression model, RF model and XGBoost model

| | 2011 summer | 2008 summer | 2011 winter-spring | 2014 winter-spring |
|---|---|---|---|---|
| Predicted burned area (km$^2$) | 2.60 | 0.18 | 1.33 | 0.37 |

18. P12L369: Can you elaborate on why the predictors explained much less of the variability of the burned area during winter-spring season compare to summer fire season?
This may be explained by the stricter fire regulations during summer in the southern states, such as Texas (While and Hanselka, 2000). For the summer fire season, under strict fire regulations, environmental factors such as high temperature or low relative humidity are relatively dominant in wildfire development. For the winter-spring fire season, more human perturbations may be involved in the control of wildfires. As the human factor in the model does not capture such perturbation, less variability was explained by the model for the winter-spring season. We have included the above discussion into the manuscript in line 391-395.

19. P13L391: Is it possible to provide information about the relationships between the amount of burned area and the most important predictors for each fire season. For example, if the relationship between RH and the burned area is negative for both the winter-spring and summer season? I would assume that moisture during the antecedent conditions should be positively correlated with the burned area while an excess of moisture during the fire season will suppress fire activity. It would be interesting to see if those relationships vary depending on the season.
The correlation between RH and burned area is -0.34 and 0.41 for winter-spring and summer fire season, respectively. This indicates that larger burned area is usually associated with the lower fire season RH relative to the past climatology for both fire seasons.

In terms of the relationship between burned area and moisture during the pre-fire season, we compared two variables that were included in both fire seasons. The correlation between burned area and the average of daily precipitation of months *t-1* is -0.05 and -0.28 for winter-spring and summer fire season and the correlation between burned area and the average of SPEI of pre-fire seasons (months of *t-1* to *t-3* for winter-spring and *t-1* to *t-2* for summer) is -0.28 and -0.34. Although lower moisture during the pre-fire season increases burned area for both fire seasons, the summer fire season has a stronger negative correlation between burned area and moisture during the pre-fire season. The differences can be explained by the key process controlling burned area for the two fire seasons. For summer, since vegetation is relatively sufficient, fuel drying in fire season and pre-fire-season is more important for wildfire development. As for winter-spring fire season, considering the vegetation amount is not as much as in the summer fire season, both fuel abundance and fuel drying in the pre-fire-season are critical for wildfires development. The balance between the two processes may explain the weaker negative correlation between burned area and moisture in the pre-fire season for the winter-spring. We have included the above discussions into the manuscript (line 458-468).

20. P14L418, L427: In "fire burned area" the word "fire" should be removed.
We have removed the word "fire".

21. P14L431: You claim that your large sample size captures the variability in wildfire activity while in L515 under limitations you admit that the model cannot predict burned area greater than it was observed before. These two sentences contradict each other. In reality, it only captures variability within 14 years which is not a long time period in terms of the climate-fire relationship. The assumption that the amount of burned area can never exceed the one that was observed in the past 14 years is flawed and cannot be held according to numerous future predictions few of which the authors cited in the introduction. While the study area is represented mostly by grassland which might have short enough fire return interval to be captured during 14 years, forested regions experience fire every 100 years and longer; therefore, this particular method cannot be used in regions with fire reoccurrence longer than a study period which needs to be clearly stated. And even in grassland, fire activity can change drastically due to climate change, population growth, social-economical changes. This limitation is too important to overcome by simply mentioning it at the end of the paper. While this model can be used to evaluate what environmental factors drive present fire activity which can benefit fire modeling, it cannot be used for future predictions.

First, given the 14-years data, assuming the fire-climate relationships are unique for each individual grid, we can have 400 (total number of grids)*14 (years)=5600 fire-climate relationships. For one grid, it can be predicted to have burned area larger than it has been observed before by learning the relationship from other grids. Thus, our model is feasible for future prediction if only considering the fire-climate relationship and assuming the effect of human activities on wildfires in the future is same as in the present day. Second, considering the majority of grids over the study domain are grassland/plain with short fire interval (~1 year), the 14-year data is suitable for assessing fire variability for our study domain. Within this 14 year period, some regions (e.g. SE Texas) experienced the largest wildfire and the most severe single year drought in the past 50 years (i.e., 2011 Texas wildfire). We have added the discussion of fire return interval and potential limitation into the paper (line 491-495).

We agree with reviewer's comment about the application of the model in terms of the short fire return interval. Our method can be applied to other regions if more data are included. For future application, more socioeconomical factors are indeed needed to be considered. We have stressed the point and included the above discussions in the manuscript (line 494-495and 597-598).

22. P14L445: Please, rewrite this sentence. The burned area is not "contributed by" any controls. We have rewritten the sentence (line 510).

23. P14L446: I do not understand how all factors can increase the burned area? I would expect that some predicted variables are negatively correlated.
To check whether or not all factors would increase the burned area, we calculate the effect of each factor in percentage by dividing the total burned area of the month, as shown in Fig R8. Note that this figure is different from Fig S9 in the manuscript whose percentage was calculated by taking the absolute value of the effects and scaling them by the number of the variables. As shown in Fig R8, generally for the months with large burned area (e.g. Jan 2006 and Sep 2011), weather, fuel, climate, and fix effect tend to increase burned area. This is consistent with the results in Fig S8. This is not the case for some months with relatively small burned area, such as Feb 2012 where the interaction (-143%), climate (-1.4%), and weather effect (-33.8%) reduce the burned area but fuel (12%) and fix effect (266%) together increase the burned area. We have revised the sentences

and included Fig R8 into the supplementary (line 512-517).

[Figure]

**Fig. R8.** Timeseries of the percentage for the (a) winter-spring fire season and (b) summer fire season. Color of blue, green, yellow, red, and purple indicate effect of weather, fuel, climate, fix, and interaction. The percentage was calculated by dividing the total burned area of the month. (This figure is now Fig. S11. In the revised manuscript)

24. P15L460: Which countries exactly when you only cite studies from the US?
To avoid confusion, we have removed the 'countries' in the sentence and added 'in the western US' (line 531).

25. P15L470-473: Add citations.
We have included the corresponding references into the paper (line 543-544).

26. P15L477: There is no doubt that climate variables will appear the main driver of fire activity considering that fires outside of natural fire seasons were excluded, together with prescribed fires to concentrate solely on environmental factors that do not include any variables other than climate.

Other environmental variables that could be included are soil, elevation, slope. But climate anomalies and climate means are both climate variables even if the authors artificially separated them into different categories.

The average of variable importance presented here have been scaled by the number of variables in the category, so the variable importance presented here is not affected by the number of variables we included in the categories. Thus, the relative importance of climate is much larger than the importance of weather and fuel, even though the number of fuel variables (5 for winter-spring and 3 for summer) and weather variables (17 for winter-spring and summer) are roughly equal and larger than the number of climate variables (6 for winter-spring and summer).

The variables suggested by the reviewer, such as topography, were considered in the process of variable selection. Topography affects the local meteorology the spread of wildfires, in particular over the regions with complex mountainous topography (Dillon et al., 2011). Since the variations of slope and elevation are relatively small over the study domain (Fig R9), we did not select them as predictors.

[Figure]

Fig. R9. (a) Elevation and (b) slope map from USGS 3D Elevation Program. Darker color indicates higher elevation (brown) and steeper slope (red). The red box denotes the selected South Central US domain.

We agree that climate anomalies, long-term mean and standard deviation of meteorology can be considered as climate variables, but in order to separate the effects of time-varying and non-time-varying climate variables, the long-term mean and standard deviation of meteorology are considered as fixed variables. We have included the reason why long-term mean and standard deviation of meteorology are considered as fixed variables in the manuscript (line 154-156 and 503-505).

27. P16L486: I would suggest either find a better way to deal with categorical variables or not including them since they are not treated equally with continuous variables.

The way we encoded categorical variable, called one-hot Encoding, is commonly-used method to encode categorical variables where there is no ordinal relationship between the categorical variables. Since there is no hierarchic relation between various land types/ecoregions, we use one-hot encoding method to encode them. Given land types and ecoregions are well known factors in wildfire development and one-hot encoding is the suitable method to encode the information, we prefer to keep the variables and the encoding method.

In general, section 5.2 is not providing any important information and I would suggest removing it from the paper. All variables are arbitrarily assigned to 4 groups while in reality all of those variables except population density are climate variables to some degree. I would suggest concentrating on specific variables which number can be significantly reduced after accounting for multicollinearity. The effect of each category on the amount of burned area is directly related to the choice of variables and is very subjective.

The weather variables and fuel variables are calculated at monthly time scale and thus should not be considered as climate variables, just like we don't call tomorrow's weather as climate. The purpose of section 5.2 is to quantify the contributions of the four environmental controls on burned area. It is different from section 5.1 which mainly focuses on the contribution of each individual variables at grid level. The variable importance was measured considering the prediction of each grid over the study period, while it cannot provide importance metrics for a specific time point or domain. The approach we used in section 5.2 was able to isolate importance of environmental controls of the burned area across the whole domain for each month. With this approach, variation of importance by months or by fire size can be quantified. We included the results and discussion of time series of importance. The details can be seen in comment #23 above.

28. P16L495: This is not Discussion and Conclusion section. It is Limitations and Conclusion. All the discussion is included together with results.

We have revised the title of this section to "Concluding remarks".

29. P16L502: I do not agree that R2 0.4 is high enough to overperform most past fire studies.

The original statement here specifically compared with prior studies predicting burned area at coarser spatial and temporal resolution. To avoid confusion, we have replaced '…outperformed most past studies that predicted wildfire burned area at a coarser spatial and temporal scale' with '…made a significant improvement to the prediction for the cases with unevenly-distributed burned area compared to past studies' (line 570-572).

30. P16L507: Here you reported the percentage of the grids with a correlation higher not "larger" than 0.5. This implies that you consider correlation 0.5 and higher significant. In this case, you should use the same threshold for multicollinearity. Or report the percentage of the grids with a correlation higher than 0.7.

We did not use the same threshold for several reasons. The threshold of 0.7 is specific to describe when collinearity begins to degrade model performance, while the correlation of 0.5 is a relatively arbitrary choice to summarize model performance as it is impossible to report every grid's correlation coefficient. Thus, the choices of the two thresholds do not need to different.  The correlation threshold used to report model performance was based on Chen et al. (2016), while the

threshold of multicollinearity was from Dormann et al. (2013). Additionally, as comment #13 shows, using the threshold of 0.5 for collinearity leads to larger bias in classifying burned grids and predicting burned area.

31. P16 L513: As I mentioned before this is a very big limitation and it needs to be discussed and suggestions for its overcoming need to be proposed. Additionally, it is necessary to state how large should be the study area to have enough data to obtain a distribution which will be representative together with how long time series is needed depending on fire regime characteristics of the study area. Failing to convince a reader that all limitations can be overcome or at least in which case the assumptions of the model are valid is the main problem with this paper. It is not clear how this technique can be transformed to a different region with higher year-to-year variability in fire activity comparing to South Central US.
See the response to comment #21 above.

32. Part of this manuscript is written in the past tense and part in the present tense. Al least within a section you should select one and be consistent.
We have changed the tense of all the paragraphs to present tense. The past tense is used when we describe previous work.

[revised manuscript text omitted]

---

## Author Response (AR4)

**Response to Reviews**

We thank the editor and the reviewers for their constructive comments to improve the manuscript. Their comments are reproduced below with our responses in blue. The corresponding changes in the manuscript are highlighted in blue.

**Editor**

While the manuscript has been improved, there are still a few major concerns need to be addressed. Please revise the manuscript according to the two additional points raised by referee #3 (attached). Also, it seems that one the major concerns about the necessity to use 58 variables (since many of which are highly correlated) from previous round has not been fully addressed. The authors may consider to use some statistical ways to test for multicollinearity, e.g., computing the variance inflation factor (or VIF), rather than choosing 0.7 as a threshold for correlation.

The issue of multicollinearity should indeed be considered more quantitatively. We stated in the original manuscript that the random forest as a machine learning tool is less unaffected by the issue of multicollinearity than traditional regression methods because the random forest randomly selects predictors used for each tree, in which the probability of sampling strongly correlated variables in a particular tree is largely avoided (Siroky, 2009). To prove this for our model, we calculate VIF for our random forest model by a bootstrapping of seven predictors (the number of predictors used in each tree) for 5000 times. We randomly select seven predictors out of all 58 potential predictors and compute the VIFs, and we repeat this sampling 5000 times for a VIF distribution. Each sampling yields seven VIFs values, and hence for 5000 sampling we obtain 35000 VIFs which forms a distribution. Figure R1 shows the distribution of VIFs for all the selected predictors. The distribution has a median of 1.67 for the winter-spring and a median of 1.62 for the summer fire season. The distribution has about 96% of the VIF values smaller than 10 for both fire seasons, demonstrating the minimized multicollinearity in tree models. We thus contend that all 58 potential predictors should be kept as model inputs and we should let the random forest algorithm choose the best predictors for itself. We have included the above discussion into the manuscript (line 503-511). Figure R1 is added in the supplementary.

[Figure]

**Figure R1**. Distributions of VIF calculated based on randomly selected seven variables of 5000 times sampling for winter-spring (top) and summer fire season (bottom) (This figure is now Fig. S12. in the revised manuscript)

In addition, the South Central US has been chosen as a study area where the risk of wildfires has been predicted to be the highest in 2031-2050. But the proposed model seems to fail to predict BA during the years with abnormal fire activity especially during spring fire season (figure 5). If it is because "random forest or quantile regression forest cannot predict burned area greater than it observes before", how would that influence the performance of the proposed model for future predictions assuming that fire activity will increase in the next several decades. Please comment on it and discuss it in the revised manuscript.

Our model is able to predict future burned area for the following two reasons. First, the predicted burned area across the whole domain for the future scenario can be larger than it has observed before. The limitation that the maximum observed burned area cannot be exceeded is applicable *only at the grid level* and this upper limit is taken from all available grids of the whole training period, which can be referred to as the global upper limit per grid. The global upper limit is 514 km$^2$ per grid for the winter-spring fire season, and 238 km$^2$ per grid for the summer fire season. Under the effect of climate change, the total burned area summed across the domain can greatly exceed the present-day total burned area. Figure R2 shows an example for a randomly selected grid box. The model can predict the largest burned area on Feb 2008 and this is consistent with the observed burned area. This demonstrates that any single grid can predict burned area larger than the grid maximum by learning from other grids, and that therefore a much larger total burned area for the domain can be predicted by our model under future climate change.

[Figure]

**Figure. R2.** Timeseries of observed (black line) and predicted total burned area (red line) for the selected grid (Lon: -98.75, Lat: 29.25) for the winter-spring fire season. (This figure is now Fig. S16. in the revised manuscript)

Second, the global upper limit is a sufficiently large value and thus the burned area per grid in the future would hardly exceed the global upper limit per grid. To further demonstrate the global upper limit per grid would be rarely exceeded, we show in Figure R3 the distribution of gridded

burned area for year 2011, an extremely severe fire year for the study domain, in comparison to the distribution of all other years for 2002-2015. It can be seen that the majority of the burned areas for the extreme year are still within the range of the observed burned area in 2002-2015. Only two grids with burned areas exceed the global upper limit from 2002-2015 (excluding 2011). The total burned area of those exceedance grids only accounts for 20% of total burned area for 2011, which is within the stated uncertainty range of our prediction model. The above discussions have been included in the manuscript (line 622-636).

[Figure]

**Figure. R3.** Distribution of burned area of all the grids for the study period excluding 2011 (black line) and of the grids for the extreme year 2011 (red line) combined both seasons. (This figure is now Fig. S17. in the revised manuscript)

**Reviewer #3**

1. Per previous suggestion, the authors give the parameter information of the XGBoost. The different parameter configurations in the XGBoost and Random Forests are used for winter-spring and summer. Perhaps, we want to a uniform robust machine learning model that can achieve high accuracy both in the different seasons.
We understand the reviewer's perspective about a unified robust model configuration. However, we have used two different sets of predictor variables for the two fire seasons to characterize different important factors and processes, because the length and characteristics of the two pre-fire seasons are fundamentally different. In this regard, using a single set of parameter configuration for two different input predictor variables could not give us two fully optimized prediction models. Two parameter configurations that are tailor-made for two separate input predictor variables are needed to fully optimize the two prediction models. Using one unified parameter configuration for both seasons can technically be achieved easily, but it is not the best approach from the perspective of fine-tuning machine learning models. We have included the above explanations in the manuscript (line 241-244).

2. The authors analyze how RH anomaly and temperature anomaly affect the prediction. But the temperature anomaly is just ranked 10th in the summer season. I think the authors need to analyze how the top at least 3 variables affect the prediction so that we can learn something from the machine learning model, not just the accuracy. On the other hand, the author said "The

physical reason behind their importance is that higher temperature coupled with lower relative humidity in the summer can cause drier fuel and this condition is favorable for fires to start, spread, and burn more intensely". But, the machine learning importance cannot provide the influence of change of variable values. The authors should further prove that.

(1) The analyses of relationship between RH anomaly, temperature anomaly, and burned area demonstrate different controls of burned area in the two fire seasons. To better understand how the changes of top variables affect burned area, the partial dependence plots can be applied to the built model and show the marginal effect of a variable on the prediction performance (Friedman, 2001), as suggested by the reviewer. As we only included the results of partial dependence plots of the top two variables for the winter-spring fire season in the manuscript, the results of other top ranked variables and for the summer fire season are similar and more discussions are provided here. Figure R4 shows the partial dependence plots for the model and the top four variables (RH anomaly, SPEI_mean4m, apcp_avg, and temp_sd) for the winter-spring fire season. For RH anomaly, the fitted logarithmic burned area is getting larger if the RH anomaly is smaller than 2% (Figure R4a). The change likely indicates the sensitivity of burned area to the fire-season moisture. Similar pattern is also shown in the partial dependence plot of the mean SPEI of the preceding 4 months (Figure R4b). Larger fitted burned area is observed to be associated with the preceding SPEI smaller than zero, suggesting that burned area in this season is highly dependent on the pre-fire-season drought conditions, which is consistent with the findings of prior studies (Scott and Burgan., 2005; Riley et al., 2013; Turco et al., 2017). As for the average precipitation of 1979-2000, the fitted burned area increases as the average precipitation increases (Figure R4c). This implies that larger fires occur in the areas where the average precipitation was more in the past. For standard deviation of temperature during 1979-2000, the fitted burned area declines dramatically when the standard deviation of temperature is larger than 9K, suggesting burned area may be larger with relatively less variation of temperature in the winter-spring fire season (Figure R4d).

[Figure]

**Figure. R4.** Partial dependence plots for the burned area model and (a) RH anomaly, (b) the mean SPEI of the preceding 4 months, (c) the average precipitation of 1979-2000, (d) the standard deviation of temperature of 1979-2000 for the winter-spring fire season. The blue line is the LOESS smooth line. (This figure is now Fig. S9. in the revised manuscript)

For the summer fire season, the large burned area is associated with low values of RH anomaly, minimum RH anomaly, the mean SPEI of the preceding 2 months, and long-term (1979–2000) standard deviation of temperature (Figure R5). The fitted logarithmic burned area increases rapidly as the RH anomaly decreases toward zero and the increase in burned area reaches a maximum at RH anomaly of –14% (Fig. R5a). Compared to the partial dependence plot for RH anomaly, the fitted burned area increases more rapidly with decreasing minimum RH anomaly (Fig. R5c). At below zero, the sensitivity of log(burned area) to the minimum RH anomaly is 0.04 %$^{-1}$ (Fig. R5c), while the corresponding sensitivity to RH anomaly is only 0.02 %$^{-1}$ (Fig. R5a). The stronger sensitivity of burned area to minimum RH anomaly indicates the stronger effects of extremely low humidity conditions on fire growth as compared with the mean RH conditions. For the standard deviation of temperature during 1979-2000, larger burned area is observed with smaller standard deviation of temperature in the past. This suggests burned area would become larger for the grids with less variation of temperature (persistent high temperature) in the summer. As for the mean SPEI of the preceding 2 months, we see an increase of fitted burned area at zero, with the largest increase at –1.8, which supports the importance of fuel drying process in the summer fire season.

[Figure]

**Figure. R5.** Partial dependence plots for the burned area model and (a) RH anomaly, (b) long-term (1979-2000) standard deviation of temperature, (c) minimum RH anomaly, and (d) the mean SPEI of the preceding 2 months for the summer season. The blue line is the LOESS smooth line. (This figure is now Fig. S10. In the revised manuscript)

For both fire seasons, RH anomaly, mean SPEI of preceding months, and standard deviation of temperature for 1979-2000 are selected as the top 4 predictors, highlighting the importance of the common variables of the two seasons but with different thresholds and magnitudes in their effects on burned area. We have included the information and the above-mentioned examples in the revised manuscript (line 447-459 and 472-486).

(2) The statement of "This highlights the importance of the stronger combined effects of RH and temperature anomalies on burned area during summer, when higher temperature coupled with lower relative humidity can cause drier fuel and create favorable conditions for fires to start, spread, and burn more intensely" is mainly based on differences in correlation between RH anomaly and temp anomaly for the two fire seasons. Additionally, in the variable importance analyses, RH anomaly is selected for both seasons, while temperature anomaly is only shown for the summer fire season. To further prove the statement, here we plot out the relationship between RH anomaly, temperature anomaly, and burned area. We perform a regression for the RH anomaly (y) and temperature anomaly (x), and fires with different sizes labeled with different

colors. The slope of the line is the change in RH anomaly over the change in temperature anomaly, which represents the dependence of RH anomaly on temperature anomaly. The slopes are -3.7 and -0.89 for the summer and winter-spring fire season, respectively, showing that a strong dependence of RH anomaly on temperature anomaly in the summer (Figure R6). In addition, large burned area (75[th] percentile, black dots in Figure R6) mainly occur in the condition of low RH anomaly and high temperature anomaly (bottom-right corner), in particular for the summer fire season. The conclusion from this plot supports our statement that "higher temperature coupled with lower relative humidity can cause drier fuel and create favorable conditions for fires to start, spread, and burn more intensely". We have revised the corresponding paragraphs and included the above discussions and Fig R6 into the supplementary (line 431-436).

[Figure]

**Figure R6**. Scatter plot of RH anomaly versus temperature anomaly for (a) winter-spring and (b) summer fire season. The blue line is the fitted regression line. The color represents different sizes of fire burned area (Green: smaller than 50[th] percentile; Red: larger than 50[th] percentile but smaller than 75[th] percentile; Black: larger than 75[th] percentile). (This figure is now Fig. S8. in the revised manuscript)

[revised manuscript text omitted]

**Step 1: Predict burned area distribution at selected percentiles**

All input data → Quantile regression forest → - Predicted burned area distribution at percentiles: 45, 55, 65, 85, 95, and 99

**Step 2: Classify unburned and burned grids**

All input data → Classification Logistic model → 0: burned area =0 ha → Predicted burned area = 0

Using criterion: Probability of burn > 0.4

1: burned area > 0 ha → Grids predicted to burn

**Step 3: Predict burned area quantiles**

Grids predicted to burn → Random forest regression model → - Predicted burned grids are grouped by predicted percentiles from RF: (39,49),(50,59),(60,69),(70,79),(80,89),(>=90)

**Step 4: Assign final predicted burned area from QRF based on predicted percentiles from RF**

- Predicted burned grids are grouped by predicted percentiles from RF: (39,49),(50,59),(60,69),(70,79),(80,89),(>=90)

- Predicted burned area distribution at percentiles: 45, 55, 65, 85, 95, and 99

- Grids in the subsets are assigned to the value at the corresponding percentiles in the distribution generated in step 1 → Final predicted burned area

**Figure 2.** Illustration of the steps in the developed model. The model includes four steps and three machine learning algorithms, including a logistic model (dark blue) classifying a grid with non-zero burned area or not, a random forest model (yellow) predicting percentiles of burned area, and a quantile regression forest (dark green) predicting conditional burned area distributions.

[Figure]

**Figure 3.** Comparison between log of observed and predicted burned area (hectare) for the (a) winter-spring and (b) summer fire season in selected years: 2011 (red, year of the largest burned area), 2008 (blue, year with burned area close to the 14-year mean of its season), and 2014 (black, year with burned area close to the 14-year mean of its season). The black line represents the line of unity and the blue line is a best fit to the data by linear regression.

[Figure]

**Figure 4.** Map of monthly mean observed and predicted burned area averaged from 2002 to 2015 for the (a) winter-spring and (b) summer fire season.

[Figure]

975 **Figure 5.** Timeseries of observed (black line) and predicted total burned area (red line) over South Central US for the (a) winter-spring and (b) summer fire season.

[Figure]

980

**Figure 6.** Relative importance of the top 14 variables presented by increase in mean square errors (%Inc.MSE) for (a) the winter-spring fire season (b) summer fire season.

[Figure]

985

**Figure 7.** The mean scaled absolute percentage of the environmental controls for the winter-spring (blue) and summer fire season (red).